# Human PC4 supports telomere stability and viability in cells utilizing the alternative lengthening of telomeres mechanism

Sara Salgado [1], Patricia L Abreu [1], Beatriz Moleirinho [1], Daniela S Guedes [1], Lee Larcombe [2,3] & Claus M Azzalin [1,4✉]

## Abstract

**Cancer cells with an activated Alternative Lengthening of Telomeres (ALT) mechanism elongate telomeres via homology-directed repair. Sustained telomeric replication stress is an essential trigger of ALT activity; however, it can lead to cell death if not properly restricted. By analyzing publicly available data from genome-wide CRISPR KO screenings, we have identified the multifunctional protein PC4 as a novel factor essential for ALT cell viability. Depletion of PC4 results in rapid ALT cell death, while telomerase-positive cells show minimal effects. PC4 depletion induces replication stress and telomere fragility primarily in ALT cells, and increases ALT activity. PC4 binds to telomeric DNA in cells, and its binding can be enhanced by telomeric replication stress. Finally, a mutant PC4 with partly impaired single stranded DNA binding activity is capable to localize to telomeres and suppress ALT activity and telomeric replication stress. We propose that PC4 supports ALT cell viability, at least partly, by averting telomere dysfunction. Further studies of PC4 interactions at ALT telomeres may hold promise for innovative therapies to eradicate ALT cancers.**

**Keywords** Alternative Lengthening of Telomeres; Cancer; PC4/Sub1; Replications Stress
**Subject Categories** Cancer; DNA Replication, Recombination & Repair

## Introduction

Telomeres are nucleic acid-protein complexes located at the ends of linear eukaryotic chromosomes. The core components of human telomeres are: (i) double-stranded (ds) 5′-TTAGGG-3′ tandem DNA repeats; (ii) shelterin, a multiprotein complex comprising the two telomeric dsDNA binding proteins TRF1 and TRF2, as well as Rap1, TIN2, TPP1, and POT1; and TERRA, a long noncoding RNA produced by RNA polymerase II (RNAPII)-mediated transcription of telomeric DNA (Azzalin et al, 2007; Nassour et al, 2021; Rivosecchi et al, 2024). Telomeric DNA shortens at each cell division, eventually leading to cellular senescence or death in the absence of mechanisms that replace the lost telomeric DNA. Cancer cells bypass this limit by activating machineries that synthesize telomeric DNA de novo (Nassour et al, 2021). While most human cancers re-elongate telomeres via reactivation of the reverse transcriptase telomerase, around 10 to 15% of cancers utilize the alternative lengthening of telomeres (ALT) mechanism (Azzalin, 2024; Carson and Flynn, 2023). ALT rates can surpass 50% in some cancers of neuroepithelial and mesenchymal origin, including sarcomas, gliomas, and pancreatic neuroendocrine tumors (Azzalin, 2024; Carson and Flynn, 2023).

ALT involves a conservative homology-directed repair pathway that elongates short telomeres through break-induced replication (BIR) (Dilley et al, 2016; Roumelioti et al, 2016). ALT BIR primarily occurs within nuclear structures referred to as ALT-associated PML bodies (APBs), which contain the scaffolding protein promyelocytic leukemia (PML), DNA repair and shelterin proteins, and telomeres from multiple chromosome ends (Dilley et al, 2016; Draskovic et al, 2009; Yeager et al, 1999; Zhang et al, 2019). Telomeric replication stress (TRS) is a major trigger of ALT activity, and consistently, a fraction of telomeres in ALT cells are bound by replication stress markers, including the RPA32 subunit phosphorylated at serine 33 (pS33) (Arora et al, 2014; Cesare et al, 2009). Because excessive replicative stress can compromise cell viability, the ALT-associated TRS can be harnessed to induce cell death. For instance, depletion of the ATPase/translocase FANCM, which suppresses ALT TRS through multiple mechanisms, leads to massive telomeric DNA damage and rapid death, specifically in ALT cells (Lu et al, 2019; Pan et al, 2019; Silva et al, 2019).

The nucleic acid-binding protein positive cofactor 4 (PC4, also known as Sub1) is a transcription co-activator conserved across various eukaryotic and prokaryotic organisms (Akimoto et al, 2014; Ge and Roeder, 1994; Henry et al, 1996; Huang et al, 2012; Knaus et al, 1996; Swaminathan et al, 2016; Werten et al, 2016). Human PC4 is a highly abundant protein of 127 amino acids (aa) that binds, in a sequence-independent manner, dsDNA through its N-terminus and melted dsDNA and single-stranded (ss) DNA through its C-terminus (Brandsen et al, 1997; Garavis and Calvo, 2017; Werten et al, 1998a;

[1]GIMM - Gulbenkian Institute for Molecular Medicine, 1649-035 Lisbon, Portugal. [2]Apexomic, Stevenage Bioscience Catalyst, Hertfordshire SG1 2FX, UK. [3]TessellateBio Ltd, Stevenage Bioscience Catalyst, Hertfordshire SG1 2FX, UK. [4]Faculty of Medicine, University of Lisbon, 1649-028 Lisbon, Portugal. ✉E-mail: claus.azzalin@gimm.pt

Werten and Moras, 2006; Werten et al, 1998b; Werten et al, 1999). PC4 serves multiple functions in general cellular transcription. In vitro, PC4 stimulates basal RNAPII-mediated transcription when present at low concentrations, while, at high concentrations, it represses transcription through ssDNA binding activity (Werten et al, 1998b; Wu and Chiang, 1998). PC4 also suppresses premature RNAPII transcription termination by interacting with the polyadenylation factor CstF-64 (Calvo and Manley, 2001), and regulates RNAPIII transcription termination and re-initiation (Wang and Roeder, 1998).

PC4 family proteins support genome stability, possibly independently of their functions in transcription regulation (Garavis and Calvo, 2017). Human PC4 and its yeast ortholog Sub1 bind to DNA G quadruplexes (G4s) in vitro, and yeast Sub1 binds to co-transcriptionally formed DNA G4s in cells (Griffin et al, 2017; Lopez et al, 2017). Sub1 deletion in a topoisomerase 1-deficient strain causes genome instability linked to co-transcriptionally formed G4 DNA, and human PC4 is sufficient to suppress those defects when expressed in the same cells (Lopez et al, 2017). Additionally, human PC4 localizes to DNA damage sites, induced by treatments with replication inhibitors, such as hydroxyurea (HU) and camptothecin, or by laser irradiation, in a manner partly dependent on transcription and ssDNA binding (Mortusewicz et al, 2016; Mortusewicz et al, 2008). PC4 depletion causes cell death when combined with HU or hypoxia treatments, and a mutant proposed to be incompetent in ssDNA binding (W89A) fails to rescue viability (Mortusewicz et al, 2016; Mortusewicz et al, 2008; Werten et al, 1998b). Furthering the complexity of its functions, PC4 physically interacts with histones H2B and H3, to maintain chromatin compaction and gene silencing, and with the tumor suppressor protein p53, to enhance its association with DNA (Batta and Kundu, 2007; Das et al, 2006; Debnath et al, 2011; Rajagopalan et al, 2009).

# Results and discussion

## PC4 is essential for the viability of ALT cells

To uncover novel factors supporting ALT cell viability, we interrogated the Project Achilles database (DepMap, version 23Q2) of the Broad Institute. The database contains gene essentiality data from genome-wide CRISPR/Cas9 loss-of-function screens performed on 1098 human cancer cell lines. Through literature surfing (Farooqi et al, 2014; Hu et al, 2021; Lu and Pickett, 2022; Mason-Osann et al, 2018; Muyas et al, 2024), we identified 14 ALT cell lines, while the majority of the remaining lines are expected to be telomerase-positive. Six of the identified ALT cell lines are derived from bone cancers, five from osteosarcomas (U2OS, Saos-2, CAL72, HuO9, and NOS-1), and one from a differentiated chondrosarcoma (CAL78). The remaining lines are derived from cancers of soft tissues (Hs 729 and SW982), central or peripheral nervous system (SK-N-FI, Daoy, and CHLA-90), breast (SK-BR-3), lung (NCI-H1299), and uterus (SK-UT-1). We isolated genes with dependency scores statistically significantly higher for the ALT lines than all remaining ones. Consistent with published work (Lu et al, 2019; Silva et al, 2019) and validating our approach, FANCM had the highest dependency score for ALT cells (Fig. 1A). Among other factors essential for ALT cell viability, we

found the FANCM-interacting partners FAAP24 and CENPX, an additional Fanconi Anemia complex component (FANCF), and the shelterin component TERF2IP/Rap1. However, we focused on PC4 because it showed the most significant dependency after FANCM (Fig. 1A). The analysis of gene effect scores for PC4 across all Project Achilles cell lines revealed that four cell lines among the ones with the highest dependency on PC4 were U2OS, Saos-2, CAL72, and NOS-1 (Fig. 1B), consistent with a previous report that PC4 is critical for osteosarcoma survival (Chetverina et al, 2023). However, these four cell lines represent only about one-third of the 13 osteosarcoma lines available in the database. It is, therefore, likely that PC4 specifically supports the viability of osteosarcomas with an activated ALT mechanism rather than osteosarcomas as a whole.

To validate our in silico analysis, we depleted PC4 using two short interfering RNAs (siRNAs) targeting different regions of PC4 mRNA (siPc1 and siPc2) in a panel of ALT (Saos-2, U2OS, CAL72, and SV40-immortalized lung fibroblasts WI-38 VA13) and telomerase-positive (HOS, osteosarcoma; HeLa, cervical carcinoma; HT1080, fibrosarcoma) cells. We also depleted PC4 in primary lung fibroblasts (HLF) using lentiviruses expressing a short hairpin RNA (shRNA) with the same sequence as siPc1 (shPc1). The chosen si/shRNAs led to at least 90% depletion of the protein across all cell lines, compared to samples transfected or infected with control si/shRNAs (si/shCt; Fig. EV1A). Cell numbers steadily diminished in PC4-depleted ALT samples over a time course of 13 days. By day 13, nearly no Saos-2 and U2OS cells and only around 10% of CAL72 and WI-38 VA13 cells were left in the plates (Fig. 1C). Some negative effects on cell numbers were also seen in siPc2-transfected HeLa and siPc1-transfected HOS cells, although to a lesser extent compared to ALT cells (Fig. 1C). However, siPc1-transfected HeLa and siPc2-transfected HOS and HT1080 cells were not majorly affected (Fig. 1C). Finally, the numbers of PC4-depleted HLF cells also decreased over time, with 30% of cells left in plate at day 15 (Fig. 1C). FACS analysis of unfixed cells stained with propidium iodide (PI) showed that PC4 depletion increased by 2 to 11-fold the number of PI-permeable Saos-2, U2OS, CAL72 and WI-38 VA13 cells, while leaving HOS, HeLa and HT1080 cells unaffected (Figs. 1D and EV1B). In HLF cells, we observed an intermediate situation with a 1.5 increase in PI-permeable, PC4-depleted cells (Fig. 1D). FACS analysis of ethanol-fixed, PI-stained cells showed that a large fraction of PC4-depleted Saos-2 cells had a sub-G1 DNA content 9 days after siRNA transfection (~20 and 44% for siPc1 and siPc2, respectively; Fig. EV1C). Because PI permeability and sub-G1 DNA content are features of cell death, these data, together with our in silico analysis, establish that PC4 depletion kills ALT cells, but not telomerase-positive cells. It appears that PC4 might also support the viability of primary fibroblasts; nonetheless, additional cell types should be sampled to verify this hypothesis.

We then tested whether the deleterious effects exerted by PC4 inactivation depend on an active ALT mechanism or on defects associated with the chosen ALT cell lines and whether telomerase could avert them. We depleted PC4 in U2OS cells with impaired ALT activity achieved through depletion of the Bloom helicase (BLM) (Sobinoff et al, 2017) and assessed cell viability. Strikingly, the number of PI-permeable cells in BLM/PC4 double-depleted cells was approximately half of that observed in cells with only PC4 depletion (Figs. 1D and EV1A). Since all ALT cells used in this

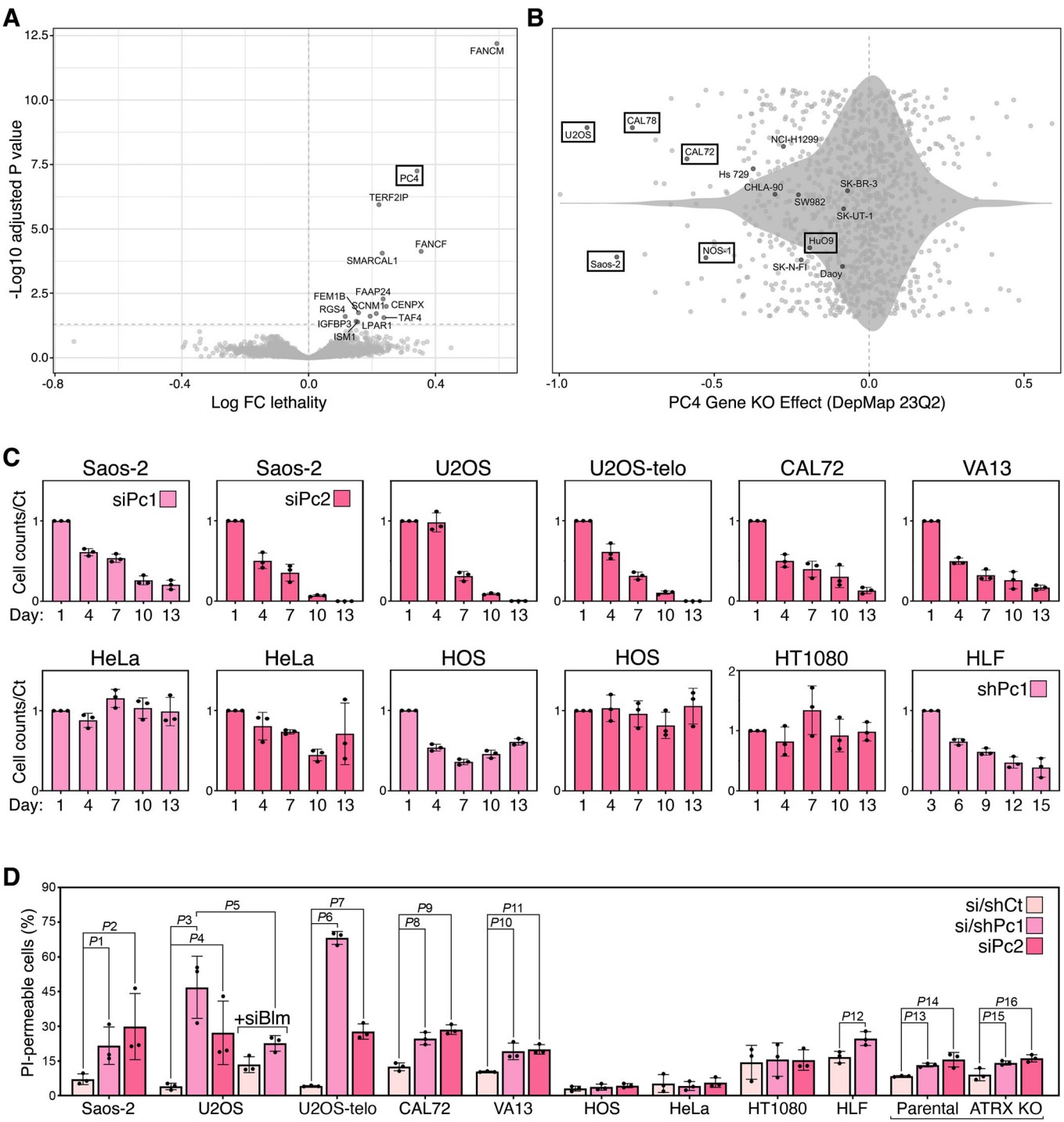

study lack the expression of the chromatin remodeler ATRX (de Nonneville et al, 2022; Lovejoy et al, 2012), PC4 dependency might result from a synthetic interaction between ATRX and PC4 deficiencies. We depleted PC4 in an ATRX-knockout (KO) HeLa S3 clone (Juhasz et al, 2018), as well as in parental HeLa S3 cells, and measured PI-permeable dead cells. Unlike parental HeLa cells, the HeLa S3 clone exhibited a 1.5-fold increase in dead cells upon PC4 depletion (Figs. 1D and EV1A), possibly due to an unknown genetic alteration. However, ATRX-KO cells behaved identically to

parental ones (Figs. 1D and EV1A). Finally, we depleted PC4 in a U2OS cell line expressing ectopic telomerase (U2OS-telo; (Silva et al, 2019)). U2OS-telo were as sensitive as parental U2OS cells to PC4 depletion, as shown by the near elimination of cells from the plates at day 13 and a sharp increase in PI-permeabilization (Figs. 1C,D and EV1A). We conclude that an active ALT mechanism, rather than ATRX loss, renders cells dependent on PC4 for survival and that telomerase expression alone is insufficient to confer resistance against PC4 loss.

**Figure 1. PC4 is essential for ALT cell viability.**

(A) Volcano plot derived from the analysis of the Project Achilles database. The plot shows the differential dependency, as Log fold change lethality, for 14 ALT cell lines versus the remaining ones. The horizontal dashed line identifies the statistical significance threshold of 0.05 (Log10 Benjamini Hochberg FDR-adjusted $P$ value). The genes in the upper right quadrant, defined by the dashed lines, are the ones with the most significant impact on ALT cells. The analysis was performed once. (B) Violin plot of the gene effect scores for PC4 across all available cell lines. ALT cell lines are indicated by their names. Boxed cell lines are derived from bone cancers. The analysis was performed once. (C) The indicated cell lines were transfected with siRNAs against PC4 (siPc1 and siPc2) over a time course of 13 days or infected with lentiviruses expressing an shRNA against PC4 (shPc1) over a time course of 15 days. Cell counts at the indicated time points are normalized against control siRNA/shRNA samples and values for the first day of counting are set to 1. Bars and error bars are means and SDs from three biological replicates. (D) Cells as in (C) were collected on day 9, stained with PI without fixation, and FACS analyzed. The fraction of PI-positive (permeable) cells is represented. siCt and shCt: control si/shRNAs; siBlm: siRNA against Bloom. Bars and error bars are means and SDs from three biological replicates. $P1 = 0.00411$; $P2 = 0.0529$; $P3 = 0.0053$; $P4 = 0.0439$; $P5 = 0.0383$; $P6 < 0.0001$; $P7 = 0.0003$; $P8 = 0.0026$; $P9 = 0.005$; $P10 = 0.0135$; $P11 = 0.0013$; $P12 = 0.0234$; $P13 = 0.0005$; $P14 = 0.0172$; $P15 = 0.0335$; $P16 = 0.0159$ (Student's $t$-test). Source data are available online for this figure.

## PC4 suppresses telomere instability and ALT activity

To test whether PC4 alleviates TRS, we depleted PC4 in several cell lines used for the viability assays and the additional ALT cell line GM847 (SV40-immortalized skin fibroblasts; Fig. EV1A) and subjected them to indirect immunofluorescence (IF) with antibodies against pS33 and, to mark telomeres, the shelterin factor TRF2. Consistent with published work (Arora et al, 2014), pS33 was readily detected at telomeres in ALT cells, even in siCt-transfected samples; moreover, PC4 depletion increased pS33 telomeric localization in all ALT cells (Figs. 2A and EV2A). PC4 depletion increased telomeric pS33 levels also in HeLa and HOS cells, but the total number of co-localization events remained consistently lower than in ALT samples (Figs. 2A and EV2A). We then performed DNA fluorescence in situ hybridization (FISH) on metaphase chromosomes to score fragile telomeres (FTs) in PC4-depleted U2OS and HOS cells. FTs are aberrant telomeric structures thought to derive from TRS, and are visualized as shredded or multiple telomeric DNA signals on the same chromatid (Sfeir et al, 2009). Consistent with ALT telomeres being more fragile than the ones of telomerase-positive cells (Arora et al, 2014; Sfeir et al, 2009), FT frequencies were higher in control U2OS than in HOS cells. Moreover, PC4 depletion increased FTs in U2OS but not HOS cells (Fig. 2B).

To evaluate the impact of PC4 depletion on ALT, we measured two ALT activity proxies, APBs and circular molecules exposing C-rich telomeric ssDNA called c-circles (Henson et al, 2009). APB frequencies were estimated by IF using antibodies against PML combined with telomeric DNA FISH. APBs were clearly detected in all ALT cells, and PC4 depletion significantly increased APB frequencies (Figs. 2C and EV2B). On the contrary, HOS and HeLa cells only showed very rare co-localization events both in control and PC4-depleted cells (Figs. 2C and EV2B). C-circles were measured using the established c-circle assay followed by dot-blot hybridization with telomeric probes (Henson et al, 2017). A 1.5-to-4-fold increase in c-circles was detected in PC4-depleted ALT cells compared to siCt-transfected cells (Fig. 2D). C-circles were not detected in any of the HOS and HeLa samples (Fig. 2D). Overall, these results demonstrate that PC4: (i) suppresses TRS in ALT cells and, possibly, also in telomerase-positive cells although to a much lower extent; and (ii) alleviates ALT activity in cells with an already established ALT mechanism, but it does not prevent ALT activation in telomerase-positive cells. We propose that PC4 alleviates ALT activity, at least in part, by restricting TRS.

Western blot analysis of total RPA32, pS33, TRF2, and PML in U2OS, Saos-2, HOS, and HeLa cells did not disclose major differences between PC4-depleted and control samples (Fig. 2E). This shows that PC4 inactivation does not cause widespread replication stress, but rather only at specific loci, including telomeres and likely other regions of unknown nature. Indeed, PC4 depletion in ALT cells also induced the appearance of pS33 foci that did not co-localize with TRF2 (Figs. 2A and EV2A). In addition, the increase in telomeric pS33 and APBs observed in PC4-depleted cells cannot be ascribed to changes in the cellular levels of the proteins visualized in our IF experiments.

## PC4 binds to telomeres in cells and its binding is increased upon replication stress

PC4 was previously identified in a proteomic study of proteins that associate with telomeres during replication (Lin et al, 2021). This, together with our results, suggests that PC4 binds to telomeres to support telomeric DNA replication and avoid TRS. We performed chromatin immunoprecipitation (ChIP) experiments using chromatin from ALT and telomerase-positive cells and a PC4 antibody, followed by dot-blot hybridization to detect telomeric or Alu-repeat DNA. Both telomeric and Alu DNAs were found in the IP fractions at comparable levels (Fig. 3A). Control ChIP experiments in U2OS cells depleted of PC4 confirmed the specificity of the antibody, as depletion nearly eliminated telomeric DNA in the IP sample (Fig. EV3A). In the same control experiments, we included U2OS and HOS cells treated with HU and found that this treatment induced a 2.5-fold increase in telomeric DNA pulled down with PC4 antibodies in U2OS but not in HOS cells (Fig. EV3A).

We then performed IF experiments using the PC4 antibody in U2OS cells transfected with siCt and siPc2. Experiments were conducted both in cells permeabilized with mild detergent treatment prior to fixation and in cells left untreated, in order to visualize insoluble and total PC4, respectively. Total PC4 produced a pan-nuclear staining while insoluble PC4 formed a large number of foci scattered across the nucleus, likely corresponding to chromatin-bound PC4 (Fig. EV3B). Both soluble and insoluble signals were greatly abolished in cells transfected with siPc2 (Fig. EV3B), confirming the specificity of the staining. Finally, we performed IF experiments to detect insoluble PC4 and TRF2 in U2OS, Saos-2, HOS, and HeLa cells depleted of FANCM, to induce TRS (Silva et al, 2019). In FANCM-proficient cells, PC4 rarely co-localized with TRF2, with only a few nuclei clearly showing at least two co-localization events (Fig. 3B,C). In FANCM-depleted ALT cells, PC4 formed large and discrete foci clearly co-localizing with TRF2, while FANCM depletion in telomerase-positive cells did not alter the co-localization frequencies (Fig. 3B,C). Altogether,

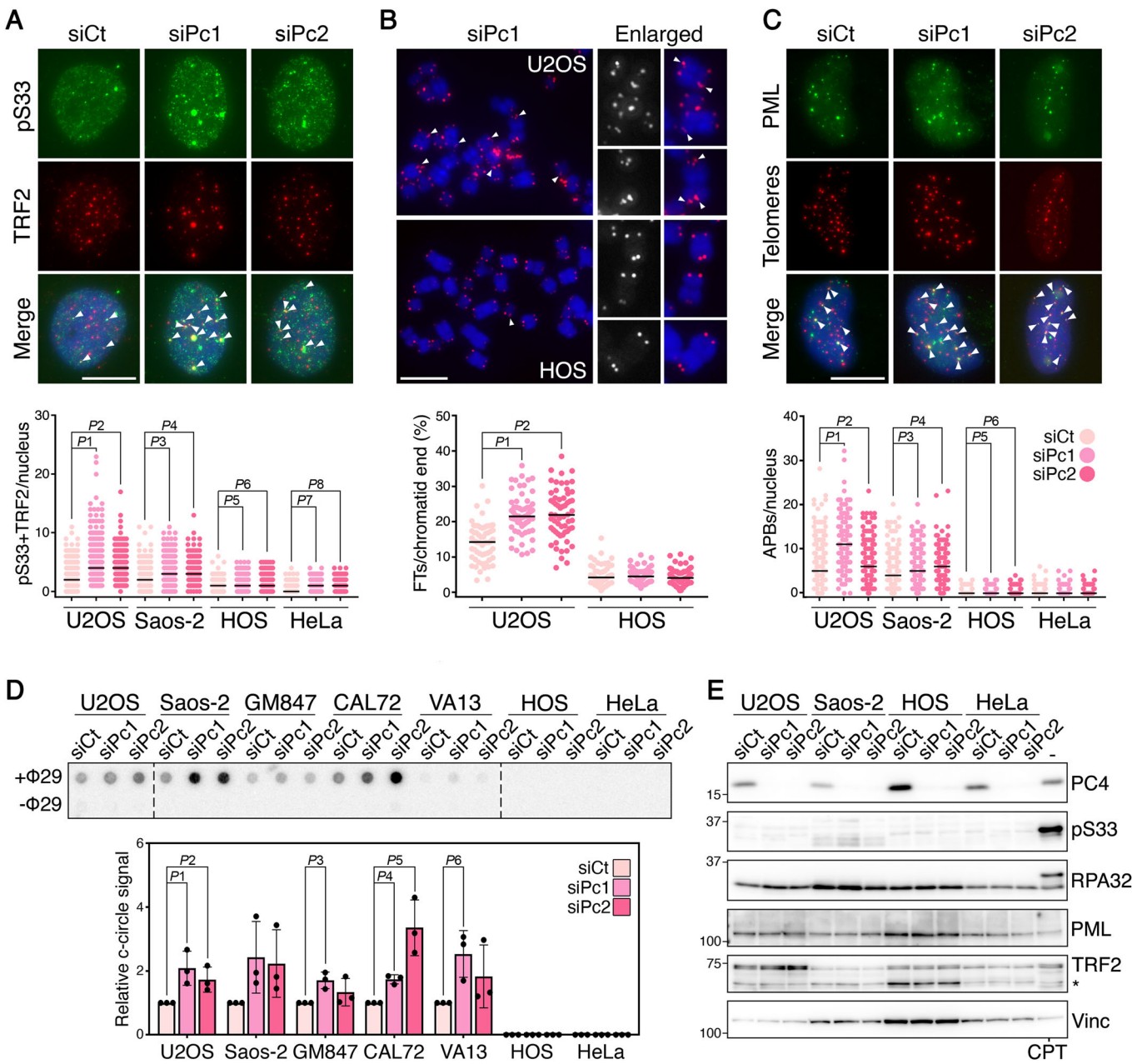

**Figure 2. PC4 restricts telomeric replication stress and ALT features.**

(A) Examples of pS33 (green) and TRF2 (red) double IF in U2OS cells transfected with the indicated siRNAs and harvested 72 h after transfection. In the merge panel, DAPI-stained DNA is in blue. Arrowheads point to co-localization events. The plot at the bottom shows the co-localization events per nucleus in U2OS, Saos-2, HOS, and HeLa cells. At least 100 nuclei were analyzed for each sample in each of the three biological replicates. $P1 < 0.0001$; $P2 < 0.0001$; $P3 < 0.0001$; $P4 < 0.0001$; $P5 < 0.0001$; $P6 < 0.0001$; $P7 < 0.0001$; $P8 < 0.0001$ (Mann–Whitney $U$-test). (B) Examples of telomeric DNA FISH on metaphases from U2OS and HOS cells as in (A). Telomeric DNA is in red and DAPI-stained chromosomal DNA is in blue. Arrowheads point to fragile telomeres (FTs). Enlarged examples are shown on the right. The plot at the bottom shows the percentage of FTs per chromatid end per metaphase. At least 20 metaphases were analyzed for each sample in each of the three biological replicates. $P1 < 0.0001$; $P2 < 0.0001$ (Mann–Whitney $U$-test). (C) Examples of PML IF (green) combined with telomeric DNA FISH (red) in U2OS cells as in (A). In the merge panel, DAPI-stained DNA is in blue. Arrowheads point to APBs. The plot at the bottom shows the numbers of APBs per nucleus in U2OS, Saos-2, HOS, and HeLa cells. At least 100 nuclei were analyzed for each sample in each of the three biological replicates. $P1 < 0.0001$; $P2 < 0.0001$; $P3 < 0.0001$; $P4 < 0.0001$; $P5 = 0.0152$; $P6 = 0.0043$ (Mann–Whitney $U$-test). In all graphs, each dot represents one nucleus or metaphase, black bars are medians. Scale bars: 10 µm. (D) Dot-blot hybridization of c-circle assay products using DNA from cells transfected with siRNAs and harvested 72 h after transfection. The graph at the bottom shows the quantification of c-circle products normalized against siCt. siCt values are set to 1. Bars and error bars are means and SDs from three biological replicates. $P1 = 0.0243$; $P2 = 0.0322$; $P3 = 0.0092$; $P4 = 0.0008$; $P5 = 0.0095$; $P6 = 0.0222$ (Student's $t$-test). (E) Western blot analysis of total proteins from cells as in (A). U2OS cells were treated with 1 µM camptothecin (CPT) for 3 h control for phosphorylated RPA32 detection. Vinculin (Vinc) serves as a loading control. Marker molecular weights are on the left in kDa. The asterisk indicates a cross-reacting band. Source data are available online for this figure.

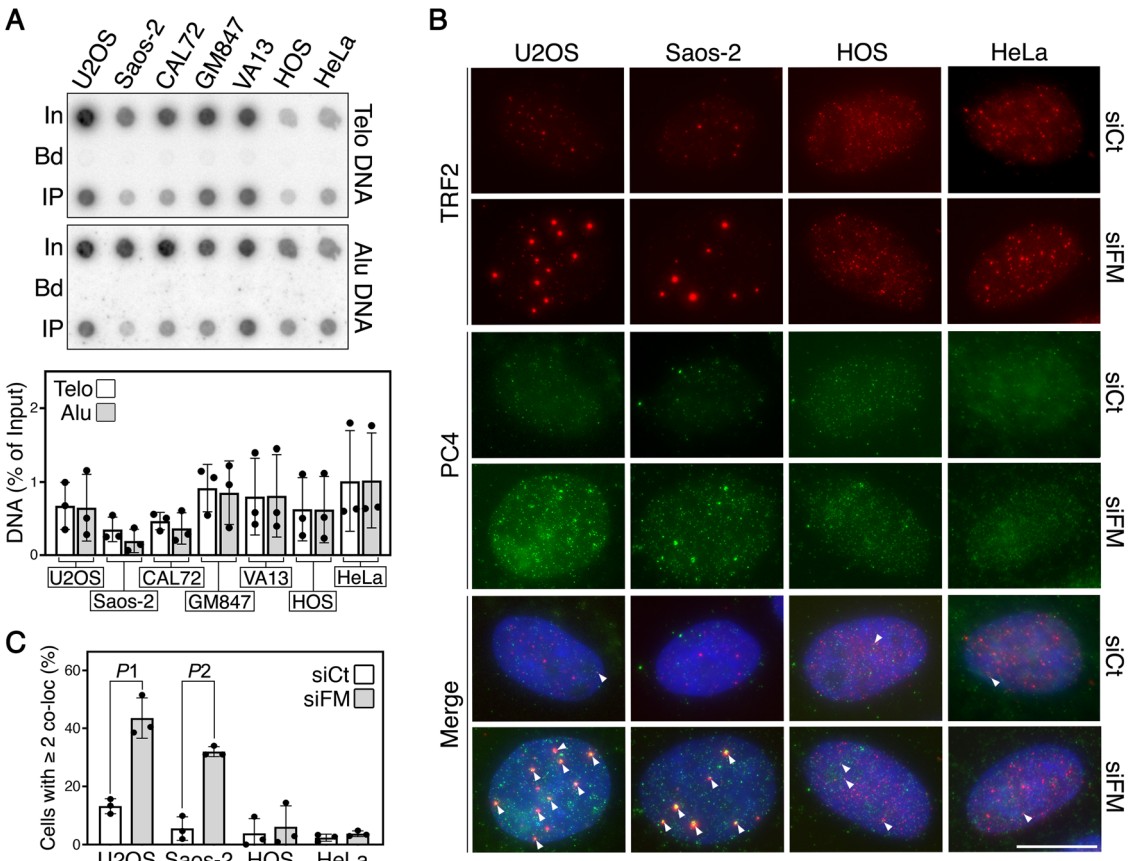

**Figure 3. PC4 associates with telomeres in cells.**

(A) Dot-blot hybridization of PC4 ChIPs using radiolabeled probes to detect telomeric (Telo) or Alu-repeat DNA. In: Input (1%), Bd: only beads control (50%), IP: PC4 immunoprecipitation (50%). Signals were quantified and graphed (bottom) as the fraction of input DNA found in the corresponding IP samples, after subtraction of Bd-associated signals. Bars and error bars are means and SDs from three biological replicates. (B) Examples of PC4 (green) and TRF2 (red) double IF in cells transfected with an siRNA-depleting FANCM (siFM) or siCt. Cells were harvested 48 h after transfection. In the merge panel, DAPI-stained DNA is in blue. Arrowheads point to co-localization events. (C) Quantifications of the percentage of cells as in (B) showing two or more co-localization events. At least 100 nuclei were analyzed for each sample in each of the three biological replicates. Bars and error bars are means and SDs. $P1 = 0.0021$; $P2 = 0.0005$ (Student's $t$-test). Scale bar: 10 μm. Source data are available online for this figure.

our ChIP and IF experiments indicate that PC4 binds to telomeres and other genomic loci in cells and that PC4 telomeric localization is enhanced by replication stress specifically in ALT cells. This strongly supports the notion that PC4 suppresses telomere instability and ALT activity in a direct manner.

## PC4 appears to support ALT telomere homeostasis through pathways that neither affect TERRA regulation nor require ssDNA binding activity

Increased RNAPII-mediated TERRA transcription and TERRA accumulation at telomeres can cause TRS (Azzalin et al, 2007; Silva et al, 2022). Moreover, polyadenylation stabilizes TERRA transcripts, conceivably expanding the pool of nuclear TERRA that can form telR-loops in trans and cause telomeric replication stress and fragility (Feretzaki et al, 2020; Porro et al, 2010). We measured RNAPII phosphorylation at CTD serine 2 and 5 (pSer2 and pSer5) in U2OS and HOS cells depleted of PC4 and detected only a minor decrease in pSer2 in PC4-depleted U2OS cells (Fig. EV4). Although this result argues against a major deregulation of TERRA production, we nonetheless performed RNA

dot-blot hybridizations using telomeric probes and did not observe major differences in total TERRA levels between control and PC4-depleted U2OS and HOS cells (Fig. 4A). We also measured TERRA degradation rates in U2OS cells transfected with siCt or siPc1 by treating cells with the transcription inhibitor actinomycin D and collecting total RNA samples every 2 h over a time course of 10 h. Although we could not extend our time course to reach TERRA half-life, as cells started to die after 10 h, TERRA decay rates were similar in siCt- and siPc1-transfected cells (Fig. 4B). Consistently, PC4 depletion in U2OS and HOS cells did not affect the steady-state levels of polyadenylated TERRA (Fig. 4C). Finally, we depleted PC4 in U2OS cells and performed TERRA RNA FISH, followed by quantification of TERRA intensity and foci number. In this case too, no major difference was observed between the control and all PC4-depleted samples (Fig. 4D). These results indicate that PC4 is not primarily involved in TERRA biogenesis or localization at telomeres, and suggest that the TRS caused by PC4 depletion does not derive from TERRA deregulation.

Because PC4 is recruited to damage sites largely through its ssDNA binding activity (Mortusewicz et al, 2016; Mortusewicz et al, 2008), such activity might be necessary for PC4 to localize to

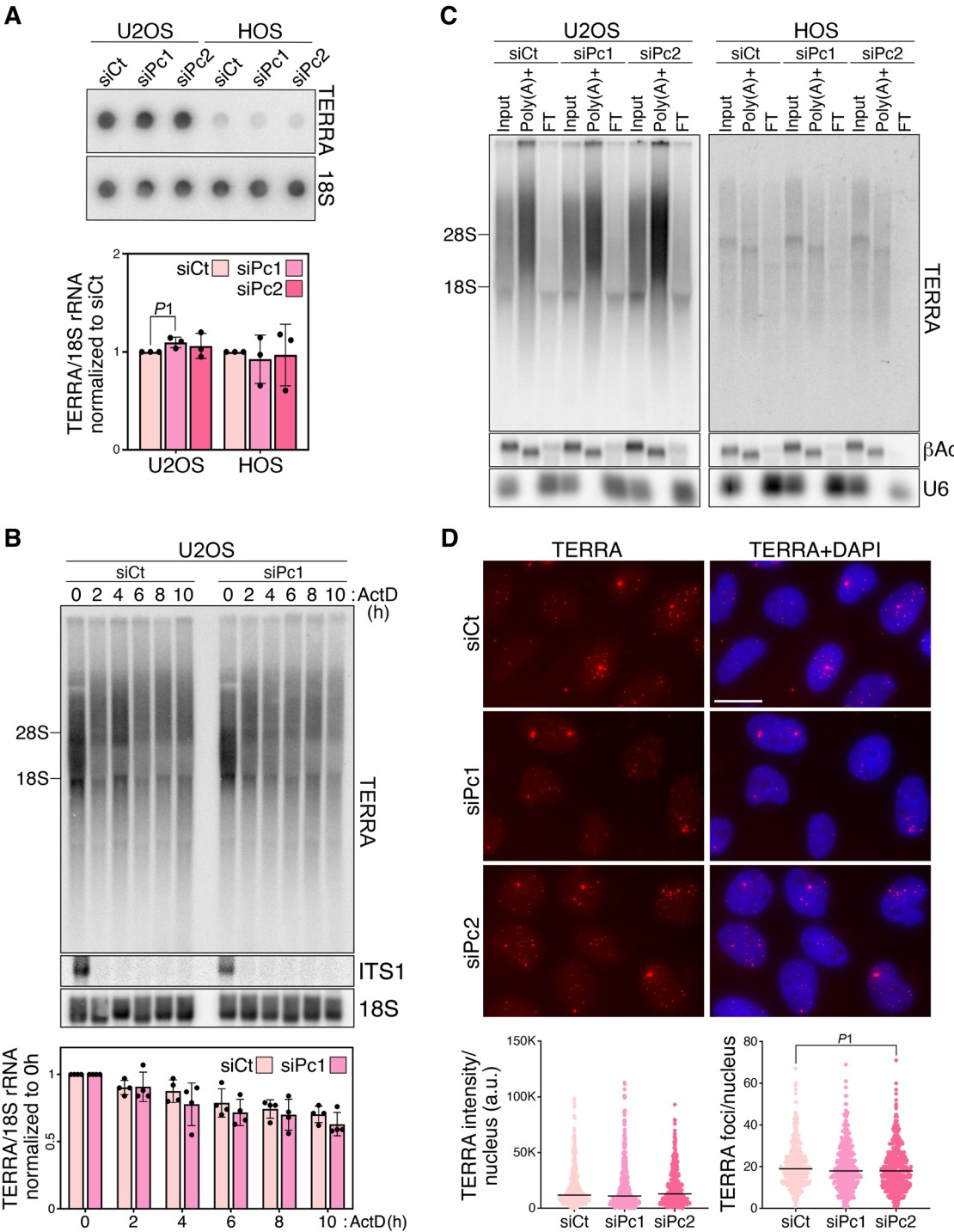

telomeres and suppress TRS. We established U2OS-derived cell lines expressing siRNA-resistant, Flag- and Myc-tagged PC4 proteins from doxycycline (dox)-inducible promoter. We generated clonal cell lines expressing either wild-type PC4 (PC4wt) or the W89A mutant and transfected them with PC4 siRNAs in the presence or absence of dox. Western blotting analysis confirmed that endogenous PC4 was efficiently depleted, while ectopic PC4wt and W89A were expressed at similar levels (Fig. 5A). The accumulation of telomeric pS33 and APBs were largely averted by PC4wt (Fig. 5B), indicating that both defects are a true outcome of

PC4 depletion. The fact that PC4wt did not fully restore pS33 and APB levels to the ones in siCt-transfected cells might derive from interference of the Flag and Myc tags with PC4 functions. Interestingly, the W89A protein was as efficient as PC4wt in preventing telomeric pS33 and APB accumulation upon PC4 depletion (Fig. 5B). We also analyzed the recruitment of PC4wt and W89A to telomeres by performing anti-Myc ChIPs in cells depleted of endogenous PC4 to avoid that the ectopic proteins could localize to telomeres through dimerization with the endogenous one. Both ectopic proteins were found at telomeres, with the W89A being

**Figure 4. Effects of PC4 depletion on TERRA biogenesis and localization.**

(A) TERRA dot-blot hybridization of total RNA from siRNA-transfected U2OS and HOS cells harvested 72 h after transfections. 18S rRNA serves as a loading control. The graph at the bottom shows quantifications of TERRA signals after normalization through 18S signals. siCt values are set to 1. Bars and error bars are means and SDs from three biological replicates. $P1 = 0.0377$ (Student's $t$-test). (B) TERRA northern blot hybridization of siRNA-transfected U2OS cells treated with actinomycin D (ActD) for the indicated hours, starting 62 h after transfections. ITS1, a short-lived precursor of the 18S rRNA, is shown to confirm the efficacy of transcription inhibition. 18S rRNA serves as a loading control. The positions of the 28S and 18S rRNAs are on indicated the left. The graph at the bottom shows quantifications of TERRA signals after normalization through 18S signals. siCt and siPc1 values at 0 h are set to 1. Bars and error bars are means and SDs from four biological replicates. (C) TERRA northern blot hybridization of input total RNA (50%), poly(A)+ fraction (100%), and flow-through (FT; 100%) from cells as in (A). Beta Actin (βAct) mRNA (poly(A) + RNA) and U6 snRNA (poly(A)- RNA) serve as loading and fractionation controls. The positions of the 28S and 18S rRNAs are indicated on the left. The image is from one representative experiment. (D) Examples of TERRA (red) RNA FISH in U2OS cells as in (A) The plots at the bottom show the quantification of TERRA intensity and foci. Each dot represents one nucleus, black bars are medians. At least 100 nuclei were analyzed for each sample in each of the three biological replicates. $P1 = 0.0526$ (Mann–Whitney $U$-test). Scale bar: 20 μm. Source data are available online for this figure.

roughly 50% less abundant than PC4wt (Fig. 5C). We then performed electrophoretic mobility shift assays with recombinant, His-tagged PC4wt and W89A purified from bacteria (Fig. EV5A) and different ssDNA oligonucleotides, including: telomeric G-rich and C-rich oligonucleotides, a mutant telomeric G-rich oligonucleotide with a G to C substitution and hence unable to form G4 structures, and an unrelated oligonucleotide. As expected from previous work (Werten et al, 1998b), PC4wt efficiently bound to all substrates (Figs. 5D and EV5B). However, unexpectedly, the W89A protein retained the ability to form complexes with all oligonucleotides, although those complexes were less stable than the ones formed by PC4wt, as shown by more smeared signals in the gels (Figs. 5D and EV5B). This last set of results can be interpreted in two different ways. It is possible that the ssDNA binding activity of PC4 is dispensable for its telomeric localization and/or functions. Alternatively, the residual ssDNA binding activity of W89A recruits enough protein to the telomeres to suppress TRS and ALT activity.

In conclusion, our research has unveiled a new suppressor of ALT activity and TRS in ALT cells and a novel factor essential for ALT cell viability. These findings point towards transient PC4 inactivation as a promising therapeutic intervention for eradicating ALT cancers, although additional experiments with multiple types of primary cells need to be conducted to evaluate the safety of those interventions. We propose that the TRS induced by PC4 depletion may contribute, at least in part, to the observed decline in cell viability. It is plausible that the impairment of other functions of PC4, such as regulating gene silencing and chromatin organization, also plays a role in cell death. Further investigations utilizing various PC4 mutants will provide insight into how PC4 sustains ALT cell viability.

## Methods

### Reagents and tools table

| Reagent/Resource | Reference or source | Identifier or catalog number |
|---|---|---|
| **Experimental models** | | |
| U2OS osteosarcoma cells (*H.sapiens*) | M. Lopes (IMCR, Zurich, Switzerland) | |
| Saos-2 osteosarcoma cells (*H.sapiens*) | B. Fuchs (Balgrist University Hospital, Zurich, Switzerland) | |
| HOS osteosarcoma cells (*H.sapiens*) | B. Fuchs (Balgrist University Hospital, Zurich, Switzerland) | |

| Reagent/Resource | Reference or source | Identifier or catalog number |
|---|---|---|
| WI-38 VA13 in vitro SV40-transformed lung fibroblasts (*H. sapiens*) | A. Londoño-Vallejo (CNRS, Paris, France) | |
| GM847 SV40-immortalized skin fibroblasts (*H. sapiens*) | A. Londoño-Vallejo (CNRS, Paris, France) | |
| HeLa cervical cancer cells (*H. sapiens*) | ATCC | Cat # CCL-2 |
| HT1080 fibrosarcoma cells (*H. sapiens*) | ATCC | Cat # CCL-121 |
| HEK293T embryonic kidney cells (*H. sapiens*) | ATCC | Cat # CRL-3216 |
| CAL72 osteosarcoma cells (*H. sapiens*) | DSMZ | Cat # ACC 439 |
| U2OS Flp-In T-Rex cells (*H. sapiens*) | U. Kutay (ETHZ, Zurich, Switzerland) | |
| Telomerase-overexpressing U2OS cells (U2OS-telo; *H. sapiens*) | Silva et al, 2019 | |
| Hela S3 cervical cancer cells (*H. sapiens*) | M. Lobrich (Darmstadt University of Technology, Germany) | |
| Hela S3 ATRX KO cervical cancer cells (*H. sapiens*) | M. Lobrich (Darmstadt University of Technology, Germany) | |
| HLF primary lung fibroblasts (*H. sapiens*) | J. Lingner (EPFL, Lausanne, Switzerland) | |
| BL21 cells (*E. coli*) | Thermo Scientific | Cat # EC0114 |
| **Recombinant DNA** | | |
| pLKO.1-puro | Sigma-Aldrich | Cat # SHC001 |
| pLKO.1-puro shRNA Control Plasmid DNA (shCt) | Sigma-Aldrich | Cat # SHC002 |
| pLKO.1-puro shPc1 | This study | |
| pMD2.G | D. Trono, EPFL, Lausanne, Switzerland | |
| pMDLg/pRRE | D. Trono, EPFL, Lausanne, Switzerland | |
| pRSV-Rev | D. Trono, EPFL, Lausanne, Switzerland | |
| pOG44 | Invitrogen | Cat # V6005-20 |
| pcDNA5-FRT-TO | Invitrogen | Cat # V6520-20 |
| pcDNA5-PC4wt-Flag-Myc | This study | |

| Reagent/Resource | Reference or source | Identifier or catalog number |
|---|---|---|
| pcDNA5-W89A-Flag-Myc | This study | |
| pET-Duet1 | Sigma-Aldrich | Cat # 71146 |
| pET-Duet1-6xHis-PC4wt | This study | |
| pET-Duet1-6xHis-W89A | This study | |
| **Antibodies** | | |
| Rabbit polyclonal anti-RPA32 pSer33 | Bethyl Laboratories | Cat # A300-246A |
| Rabbit polyclonal anti-PC4 | Bethyl Laboratories | Cat # A301-161A-M |
| Rabbit polyclonal anti-PML | Bethyl Laboratories | Cat # A301-168A |
| Rabbit polyclonal anti-RPA32 | Bethyl Laboratories | Cat # A300-244A |
| Rabbit polyclonal anti-Bloom | Bethyl Laboratories | Cat # A300-110A |
| Rabbit polyclonal anti-ATRX | Bethyl Laboratories | Cat # A301-045A |
| Rabbit polyclonal anti-RNA PolII | Bethyl Laboratories | Cat # A300-653A |
| Rabbit polyclonal anti-phospho RNA PolII (S2) | Bethyl Laboratories | Cat # A300-654A |
| Rabbit polyclonal anti-phospho RNA PolII (S5) | Bethyl Laboratories | Cat # A300-655A |
| HRP-conjugated goat anti-mouse IgGs | Novus Biologicals | Cat # NB7539 |
| HRP-conjugated goat anti-rabbit IgGs | Novus Biologicals | Cat # NB7160 |
| Rabbit polyclonal anti-TRF2 | Novus Biologicals | Cat # NB110-57130 |
| Mouse monoclonal anti-TRF2 | Sigma-Aldrich | Cat # 05-521 |
| Alexa Fluor 488-conjugated donkey anti-rabbit IgGs | Invitrogen | Cat # A21206 |
| Mouse monoclonal anti-Myc | Cell Signaling Technology | Cat # 2276 |
| Rabbit monoclonal anti-Vinculin | Cell Signaling Technology | Cat # 13901 |
| Alexa Fluor 568-conjugated donkey anti-mouse IgGs | Invitrogen | Cat # A10037 |
| Mouse monoclonal anti-beta Actin | Abcam | Cat # ab8224 |
| Goat polyclonal anti-Helic2 | Santa Cruz Biotechnology | Cat # SC-68563 |
| **Oligonucleotides and other sequence-based reagents** | | |
| Nonsilencing, negative control DsiRNA (siCt) | Integrated DNA Technologies | Cat # 51-01-14-04 |
| siPc1 | Integrated DNA Technologies (this study) | |
| siPc2 | Integrated DNA Technologies (this study) | |
| siFM | Integrated DNA Technologies (Silva et al, 2019) | |
| siBlm | Integrated DNA Technologies (Silva et al, 2019) | |

| Reagent/Resource | Reference or source | Identifier or catalog number |
|---|---|---|
| Telo2 dsDNA probe | Azzalin et al, 2007 | |
| βActin DNA oligonucleotide | Microsynth (this study) | |
| U6 DNA oligonucleotide | Microsynth (this study) | |
| 18S DNA oligonucleotide | Microsynth (this study) | |
| G-rich telomeric DNA oligonucleotide | Microsynth (this study) | |
| Alu DNA oligonucleotide | Microsynth (this study) | |
| EMSA DNA oligonucleotides | Microsynth and Integrated DNA Technologies (this study) | |
| ITS1 DNA oligonucleotide | Integrated DNA Technologies (Rouquette et al, 2005) | |
| AF568-conjugated C-rich telomeric PNA probe | Panagene | |
| **Chemicals, enzymes, and other reagents** | | |
| High glucose DMEM, GlutaMAX | Gibco | Cat # 61965059 |
| High glucose DMEM/F12, GlutaMAX | Gibco | Cat # 31331028 |
| Penicillin-streptomycin | Gibco | Cat # 15070063 |
| Hydroxyurea | Sigma-Aldrich | Cat # H8627 |
| Actinomycin D | Sigma-Aldrich | Cat # 114666 |
| Puromycin | Sigma-Aldrich | Cat # P8833 |
| Propidium iodide | Sigma-Aldrich | Cat # P4864 |
| Colchicine | Sigma-Aldrich | Cat # C9754 |
| Doxycycline | Sigma-Aldrich | Cat # D5207 |
| Formaldehyde | Sigma-Aldrich | Cat # 47608 |
| Pepsin | Sigma-Aldrich | Cat # 77151 |
| Tris | Sigma-Aldrich | Cat # T1503 |
| HCl | Sigma-Aldrich | Cat # 258148 |
| NaCl | Sigma-Aldrich | Cat # S3014 |
| Tween-20 | Sigma-Aldrich | Cat # P2287 |
| DAPI | Sigma-Aldrich | Cat # D9542 |
| $MgCl_2$ | Sigma-Aldrich | Cat # M2670 |
| PIPES | Sigma-Aldrich | Cat # 80635 |
| Triton-X | Sigma-Aldrich | Cat # T8787 |
| SDS | Sigma-Aldrich | Cat # 75746 |
| Glycerol | Sigma-Aldrich | Cat # G5516 |
| 2-mercaptoethanol | Sigma-Aldrich | Cat # 63689 |
| Bromophenol blue | Sigma-Aldrich | Cat # B5525 |
| Phenol | Sigma-Aldrich | Cat # P4557 |
| Chloroform | Sigma-Aldrich | Cat # 366927 |
| Dextran sulfate | Sigma-Aldrich | Cat # D8906 |
| EDTA | Sigma-Aldrich | Cat # E4884 |
| HEPES | Sigma-Aldrich | Cat # H4034 |
| NaOH | Sigma-Aldrich | Cat # 567530 |
| NaHCO3 | Sigma-Aldrich | Cat # 71627 |

| Reagent/Resource | Reference or source | Identifier or catalog number |
|---|---|---|
| His-tag protease inhibitors | Sigma-Aldrich | Cat # S8830 |
| dATP | Thermo Scientific | Cat # R0141 |
| dTTP | Thermo Scientific | Cat # R0171 |
| dGTP | Thermo Scientific | Cat # R0161 |
| Agarose resin charged with divalent cobalt | Thermo Scientific | Cat # 89964 |
| DTT | Thermo Scientific | Cat # R0861 |
| Lipofectamine RNAiMAX | Invitrogen | Cat # 13778150 |
| TRIzol | Invitrogen | Cat #15596018 |
| Dynabeads Protein G | Invitrogen | Cat # 10004D |
| Lipofectamine 2000 | Invitrogen | Cat # 11668019 |
| Ethanol | Supelco | Cat # 1009832500 |
| KCl | Supelco | Cat # 1049365000 |
| Methanol | Supelco | Cat # 1060092500 |
| Acetic acid | Supelco | Cat # 1000632500 |
| BSA | New England Biolabs | Cat # B9000S |
| Oligo d(T)25 Magnetic Beads | New England Biolabs | Cat #S1419S |
| T4 polynucleotide kinase | New England Biolabs | Cat #M0201L |
| Phi29 DNA polymerase | New England Biolabs | Cat # M0269 |
| Vanadyl Ribonucleoside Complex | New England Biolabs | Cat # S1402S |
| Amersham Protran 0.45 NC | Cytiva | Cat #10600002 |
| ECL detection reagents | Cytiva | Cat #RPN2236 |
| Amersham HybondTM-N+ | Cytiva | Cat #RPN203B |
| RNaseA | NZYtech | Cat # MB18701 |
| BSA | NZYtech | Cat # MB04602 |
| Bradford reagent | NZYtech | Cat # MB19801 |
| BlueSafe | NZYtech | Cat # MB15201 |
| Formamide | PanReac | Cat # A2156.1000 |
| IPTG | PanReac | Cat # A1008,0025 |
| Imidazole | PanReac | Cat # A1073.500 |
| PMSF | Roche | Cat # 10837091001 |
| COmplete Protease Inhibitor Cocktail | Roche | Cat # 4693159001 |
| Blocking solution | Roche | Cat # 11096176001 |
| [γ-32P]ATP | PerkinElmer | Cat # BLU002A250UC |
| Cy3-dCTP | PerkinElmer | Cat # NEL576001EA |
| Glycine | VWR | Cat # 103 |
| Trisodium citrate | VWR | Cat # 27833.294 |
| Protein A-Agarose beads | Santa Cruz Biotechnology | Cat # sc-2001 |
| Protein G-Agarose beads | Santa Cruz Biotechnology | Cat # sc-2002 |
| Tetracycline-free fetal bovine serum | Pan BioTech | Cat # P30-3602 |
| Camptothecin | Selleckchem | Cat # S1288 |
| Hygromycin B | Corning | Cat # 30-240-CR |
| Vectashield | Vectorlabs | Cat # H-1000 |

| Reagent/Resource | Reference or source | Identifier or catalog number |
|---|---|---|
| Sucrose | Millipore | Cat # 84100 |
| DNaseI | Qiagen | Cat #50979254 |
| His-tagged PC4wt | This study | |
| His-tagged W89A | This study | |
| **Software** | | |
| DepMap project v23Q2 | https://www.depmap.org | |
| R Statistical Software v4.3.1 | https://www.r-project.org/ | |
| ggplot2 package v3.4.4 | https://ggplot2.tidyverse.org | |
| limma Bioconductor package v3.56.2 | https://bioconductor.org/packages/limma | |
| FlowJo v10.10.0 | https://www.flowjo.com/ | |
| ImageJ | https://imagej.net/software/fiji/ | |
| Photoshop | https://www.adobe.com/pt/products/photoshop.html | |
| GraphPad Prism v8.4.3 | https://www.graphpad.com/ | |
| **Other** | | |
| Amersham 680 RGB Imager | Cytiva | |
| Amersham Typhoon IP imager | Cytiva | |
| Trans-Blot SD Semi-Dry Transfer Cell | Bio-Rad | |
| Bioruptor Plus | Diagenode | |
| MSE Soniprep 150 | Sanyo | |
| BD Accuri C6 | BD Biosciences | |
| Zeiss Cell Observer | ZEISS Microscopy | |
| Nanodrop 2000 | Thermo Scientific | |
| Gravity-flow polypropylene columns | Thermo Scientific | Cat # 29924 |
| Wizard SV gel and PCR Clean-up system | Promega | Cat # A9282M |
| Oligo Clean and Concentrator kit | Zymo Research | Cat # D4060 |
| Nucleospin RNA kit | Macherey-Nagel | Cat # 12373368 |
| LookOut Mycoplasma PCR Detection kit | Sigma-Aldrich | Cat # MP0035 |

## Project Achilles mining

Data describing cell line sensitivity to CRIPSR knock-outs was retrieved from the Project Achilles database hosted at the DepMap project (https://www.depmap.org; version 23Q2) of the Broad Institue. DepMap project files were downloaded containing the Chronos gene effect scores (CRISPRGeneEffect.csv) and the cell line annotation for all cell lines included in the project (CellLineAnnotation.csv). These were combined to generate a file containing basic cell annotation (including the common cell line

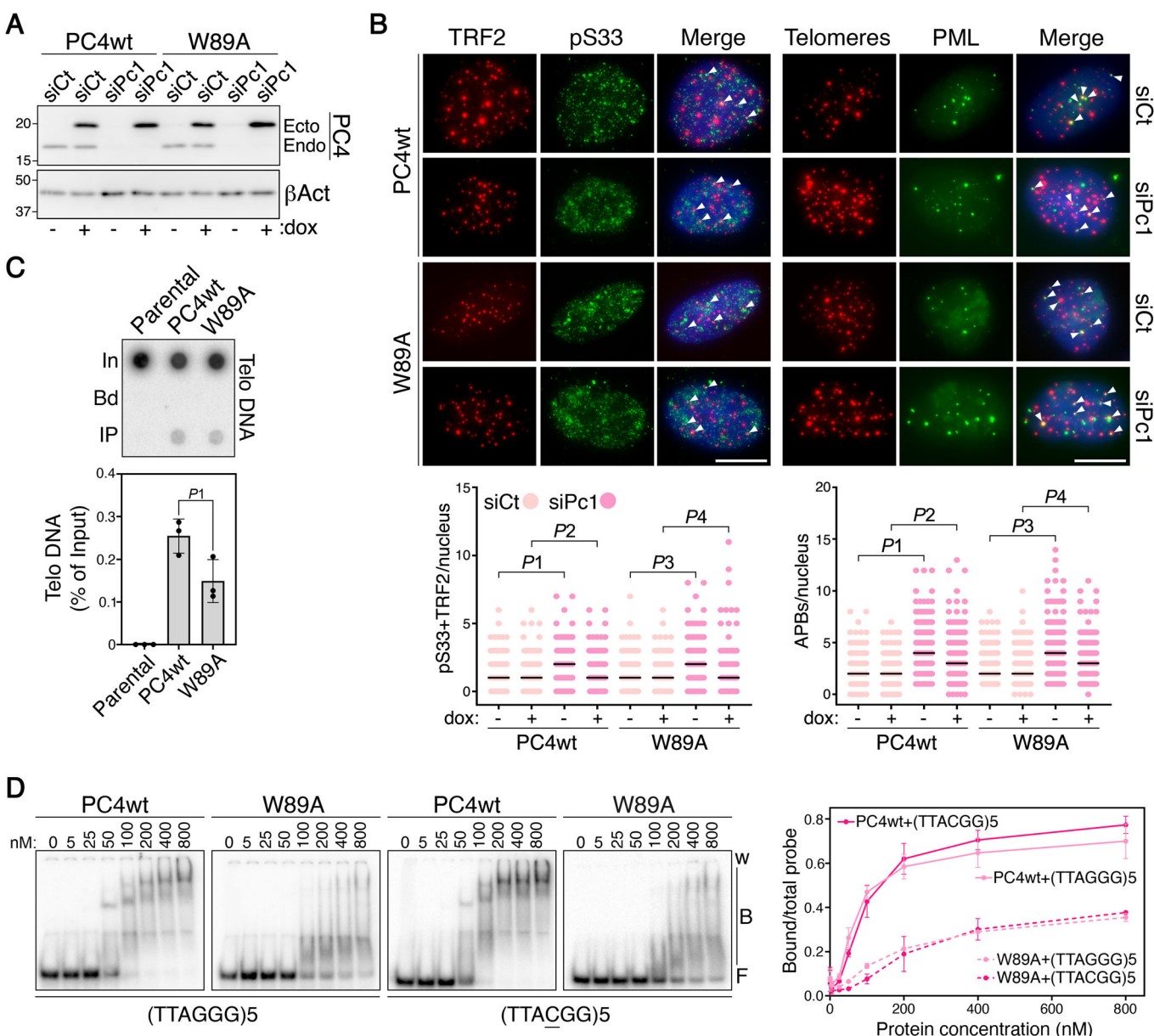

**Figure 5. The ssDNA binding-deficient mutant W89A suppresses ALT activity and telomeric replication stress and is recruited to telomeres.**

(A) Western blot analysis of endogenous (Endo) and ectopic (Ecto) PC4 proteins in U2OS cell lines expressing siRNA-resistant PC4 proteins, either wt or W89A. Cells were grown in the presence or absence of doxycycline (dox), transfected with siRNAs, and harvested 72 h after transfection. Ectopic PC4 variants run slower because they carry C-terminal Myc and Flag tags. Beta Actin (βAct) serves as a loading control. Marker molecular weights are on the left in kDa. (B) Examples of pS33 (green) and TRF2 (red) double IF (left panels) and of PML IF (green) combined with telomeric DNA FISH (red; right panels) in cells as in (A). Only cells treated with dox are shown. Arrowheads point to co-localization events. The plots show the co-localization events in experiments as above. Each dot represents one nucleus, black bars are medians. At least 100 nuclei were analyzed for each sample in each of the three biological replicates. For the plot on the left: P1 < 0.0001; P2 = 0.031; P3 < 0.0001; P4 = 0.0002; for the plot on the right: P1 < 0.0001; P2 < 0.0001; P3 < 0.0001; P4 < 0.0001 (Mann–Whitney U-test). Scale bars: 10 μm. (C) Dot-blot hybridization of anti-Myc ChIPs in cells as in (A) using radiolabeled probes to detect telomeric (Telo) DNA. In: Input (2%), Bd: only beads control (50%), IP: Myc immunoprecipitation (50%). Signals were quantified and graphed (bottom) as the fraction of input DNA found in the corresponding IP samples, after subtraction of Bd-associated signals. Bars and error bars are means and SDs from three biological replicates. P1 = 0.0470 (Student's t-test). (D) Electrophoretic mobility shift assays with recombinant PC4wt and W89A proteins and ssDNA oligonucleotides. The underlined C in the oligonucleotide on the right eliminates the formation of G4 structures. w: wells; B: bound probe; F: free probe. The graph on the right shows quantifications of bound oligonucleotides graphed as a fraction of the total signal within each lane. Data points and error bars are means and SDs from three biological replicates. Source data are available online for this figure.

names) along with the CRISPR KO effect scores and subsequently annotated with an indication of the ALT status of the cell lines, based on literature surfing (Farooqi et al, 2014; Hu et al, 2021; Lu and Pickett, 2022; Mason-Osann et al, 2018; Muyas et al, 2024). The resulting data contained details for 1098 cell lines and the CRISPR KO scores of 17,931 genes. All analyses were performed using R Statistical Software (v4.3.1 https://www.r-project.org/) and plotted using the ggplot2 package (version 3.4.4). An investigation into the differential gene KO effect between ALT and non-ALT cell lines was carried out using the limma Bioconductor package (version 3.56.2). A significant impact on ALT cell viability was assigned to genes showing $P < 0.05$ after Benjamini Hochberg FDR correction. To explore the distribution of PC4 KO effect on all cell lines, the DepMap annotated data was used to extract only gene effect scores for PC4, along with cell line annotation, for plotting in R.

## Cell lines and culture conditions

U2OS osteosarcoma cells were a kind gift from M. Lopes (IMCR, Zurich, Switzerland). Saos-2 and HOS osteosarcoma cells were a kind gift from B. Fuchs (Balgrist University Hospital, Zurich, Switzerland). WI-38 VA13 in vitro SV40-transformed lung fibroblasts and GM847 SV40-immortalized skin fibroblasts were a kind gift from A. Londoño-Vallejo (CNRS, Paris, France). HeLa cervical cancer, HT1080 fibrosarcoma, and HEK293T embryonic kidney cells were purchased from ATCC. CAL72 osteosarcoma cells were purchased from DSMZ. U2OS Flp-In T-REx were a kind gift from U. Kutay (ETHZ, Zurich, Switzerland). Telomerase-overexpressing U2OS cells (U2OS-telo) were generated and validated previously (Silva et al, 2019). Hela S3 and Hela S3 ATRX KO were a kind gift from M. Lobrich (Darmstadt University of Technology, Germany). HLF primary lung fibroblasts from a healthy female donor were a kind gift from J. Lingner (EPFL, Lausanne, Switzerland). U2OS and derivatives, HeLa and derivatives, HOS, HT1080, and HLF cells were cultured in high glucose DMEM, GlutaMAX (Gibco) supplemented with 5–10% tetracycline-free fetal bovine serum (Pan BioTech) and 100 U/ml penicillin-streptomycin (Gibco). Saos-2, WI-38 VA13, GM847, and CAL72 cells were cultured in high glucose DMEM/F12, GlutaMAX (Gibco), supplemented with 10% tetracycline-free fetal bovine serum and 100 U/ml penicillin-streptomycin. When indicated, cells were treated with 1 μM camptothecin (Selleckchem) for 3 h, 0.2 mM hydroxyurea (Sigma-Aldrich) for 16 h, or 5 μg/ml actino-mycin D (Sigma-Aldrich) for up to 10 h. Cells were maintained mycoplasma-free and regularly checked for contaminations using the LookOut Mycoplasma PCR Detection kit (Sigma-Aldrich).

## siRNA-mediated depletion

DsiRNAs (Integrated DNA Technologies) were transfected into cells using the Lipofectamine RNAiMAX reagent (Invitrogen) at a final concentration of 20–30 nM. The transfection medium was discarded and replaced with a normal culture medium 5 h after transfection. DsiRNA target sequences were as follows: siPc1: 5′-GAACAGAUUUCUGACAUUGAU-3′; siPc2: 5′-UGACAUU-GAUGAUGCAGUAAG-3′; siFM: 5′-GGAUGUUUAGGAGAA-CAAAGAGCUA-3′ (Silva et al, 2019); siBlm: 5′-GCTAGGAGTCTGCGTGCGAGGATTA-3′ (Silva et al, 2019).

The siCt was a non-targeting control DsiRNA (Integrated DNA Technologies).

## shRNA-mediated depletion

Lentiviral particles were produced by co-transfecting HEK293T cells with the pLKO.1-puro vector (Sigma-Aldrich) containing the shRNA sequences of interest, the pMD2.G envelope plasmid, and the pMDLg/pRRE and pRSV-Rev packaging plasmids (kind gifts from D. Trono, EPFL, Lausanne, Switzerland) using the calcium phosphate method. The culture medium was collected 36 h post-transfection, filtered through a 0.45 μm filter, and used to infect twice HLF cells. Cells were selected 24 h after the second infection with 1 μg/ml puromycin (Sigma-Aldrich) for 5 days. The shRNA target sequences were as follows: shPc1: 5′-GAACA-GAUUUCUGACAUUGAU-3′ (same as siPc1); shCt: 5′- CAA-CAAGATGAAGAGCACCAA-3′ (SHC002, Sigma-Aldrich).

## Generation of PC4-expressing cell lines

Wild type and W89A mutant PC4 cDNAs were synthesized at Integrated DNA Technologies. The cDNAs contain C-terminal Flag and Myc tags in frame with PC4 and silent mutations that render them resistant to both siPc1 and siPc2. The cDNAs were cloned into the pcDNA5-FRT-TO vector (Invitrogen), and plasmids were transfected into U2OS Flp-In T-REx cells together with the plasmid pOG44 (Invitrogen) expressing the Flp recombinase using the Lipofectamine 2000 reagent (Invitrogen). Cells were plated at low density in a medium containing 200 μg/ml hygromycin B (Corning), and single clones were manually isolated after 2 weeks and expanded. Clones were tested for dox-induced PC4 expression by western blotting and indirect immunofluorescence using anti-PC4 antibodies. Two independent cell lines for PC4wt (clone 6 and clone 13) and one for W89A (clone 4) were chosen for further experiments based on the homogeneity of ectopic protein expression. For induction of ectopic PC4 proteins, 25 ng/ml doxycycline (Sigma-Aldrich) was added to the culture medium 24 h before siRNA transfection and maintained until the end of the assays. PC4wt clone 6 was used for ChIPs, clone 13 was used for IF.

## Cell proliferation and viability assays

For growth curves, $1–1.5 \times 10^5$ cells were seeded in 6 cm dishes, counted, and passaged every 72 h until the end of the assays. For siRNA-mediated depletion, cells were re-transfected with siRNAs every 72 h. For fluorescence-activated cell sorting (FACS), cells were trypsinized and pelleted by centrifugation at $500 \times g$ at 4 °C for 5 min. Cell pellets were either left untreated for viability assays or fixed in 70% ethanol (Supelco) at −20 °C for 30 min and treated with 25 μg/ml RNaseA (NZYtech) in 1× PBS at 37 °C for 20 min for cell cycle analysis. Cells were washed with 1× PBS and stained with 20 μg/ml propidium iodide (Sigma-Aldrich) in 1× PBS at 4 °C for 10 min. Flow cytometry was performed on a BD Accuri C6 (BD Biosciences). Data were analyzed using FlowJo software.

## Metaphase fluorescence in situ hybridization (FISH)

Cells were incubated with 200 ng/ml colchicine (Sigma-Aldrich) for 2–6 h, and mitotic cells were harvested by manual shake-off and

centrifugation for 10 min at 400 × g, at 10 °C. Cell pellets were incubated with 0.075 M KCl (Supelco) at 37 °C for 10 min, followed by centrifugation and fixation of cell pellets with ice-cold methanol/acetic acid (3:1) (Supelco). Chromosomes were spread on glass slides, treated with 20 µg/ml RNaseA in 1× PBS at 37 °C for 1 h, fixed in 4% formaldehyde (Sigma-Aldrich) in 1× PBS for 2 min, and treated with 70 µg/ml pepsin (Sigma-Aldrich) in 2 mM glycine, pH 2 (VWR) at 37 °C for 5 min. Slides were fixed again with 4% formaldehyde in 1× PBS for 2 min, incubated with 70, 90, and 100% ethanol for 5 min each, and air-dried. 50 µl of hybridization solution (10 mM Tris-HCl (Sigma-Aldrich) pH 7.2, 70% formamide (PanReac), 0.5% blocking solution (Roche)) containing 10 nM AF568-conjugated C-rich telomeric PNA probe (TelC-AF568; 5′-AF568-OO-CCCTAACCCTAACCCTAA-3′; Panagene) were applied onto the slides followed by incubation at 80 °C for 5 min and at room temperature for 2 h. Slides were washed twice with 10 mM Tris-HCl pH 7.2, 70% formamide, 0.1% BSA (NZYtech) and three times with 0.1 M Tris-HCl pH 7.2, 0.15 M NaCl, 0.08% Tween-20 (Sigma-Aldrich) at room temperature for 10 min each. DNA was counterstained with 100 ng/ml DAPI (Sigma-Aldrich) in 1× PBS, and slides were mounted in Vectashield (Vectorlabs). Images were acquired with a Zeiss Cell Observer equipped with a cooled Axiocam 506 m camera and a 63X/1.4NA oil DIC M27 PlanApo N objective. Image analysis was performed using ImageJ and Photoshop software.

## Indirect immunofluorescence (IF)

To detect total proteins, cells grown on coverslips were fixed with 4% formaldehyde in 1× PBS for 10 min, followed by permeabilization in CSK buffer (100 mM NaCl, 300 mM sucrose, 3 mM MgCl$_2$, 10 mM PIPES pH 6.8, 0.5% Triton-X; Sigma-Aldrich and Millipore) for 7 min at room temperature. For pre-extraction of soluble material, coverslips were incubated with CSK buffer for 7 min on ice, fixed with 4% formaldehyde in 1× PBS for 10 min, and re-permeabilized with CSK buffer for 5 min at room temperature. All successive steps were performed at room temperature. Samples were incubated with blocking solution (0.5% BSA, 0.1% Tween-20 in 1× PBS) for 1 h, followed by incubation with primary antibody diluted in blocking solution for 1 h. Samples were washed three times with 0.1% Tween-20 in 1× PBS for 10 min each, incubated with secondary antibodies diluted in blocking solution for 1 h, and washed twice with 0.1% Tween-20 in 1× PBS for 10 min each. DNA was counterstained with 100 ng/ml DAPI in 1× PBS. For combined IF and DNA FISH, cells were again fixed after the secondary antibody was washed with 4% formaldehyde in 1× PBS for 10 min, washed three times with 1× PBS, incubated with 10 mM Tris-HCl pH 7.2 for 5 min, denatured and hybridized with 10 nM of TelC-AF568 PNA probe. DNA was counterstained with 100 ng/ml DAPI in 0.1 M Tris-HCl pH 7.2, 0.15 M NaCl, 0.08% Tween-20. Coverslips were mounted on slides in Vectashield. Primary antibodies were: a rabbit polyclonal anti-RPA32 pSer33 (Bethyl Laboratories, A300-246A; 1:1000 dilution), a rabbit polyclonal anti-PC4 (Bethyl Laboratories, A301-161A-M; 1:100 dilution), a mouse monoclonal anti-TRF2 (Sigma-Aldrich, 05-521; 1:500 dilution), and a rabbit polyclonal anti-PML (Bethyl Laboratories, A301-168A; 1:500 dilution). Secondary antibodies were: Alexa Fluor 488-conjugated donkey anti-rabbit IgGs (Invitrogen, A21206; 1:2000 dilution) and Alexa Fluor 568-conjugated donkey anti-mouse IgGs

(Invitrogen, A10037; 1:2000 dilution). Images were acquired with a Zeiss Cell Observer equipped with a cooled Axiocam 506 m camera and a 63X/1.4NA oil DIC M27 PlanApo N objective. Image analysis was performed using ImageJ and Photoshop software.

## Western blotting

Cells were lysed in a plate with 2x Laemmli buffer (4% SDS, 20% glycerol, 120 mM Tris-HCl pH 6.8; all from Sigma-Aldrich) and harvested by scraping. Lysates were incubated at 95 °C for 5 min, and cell debris was eliminated by high-speed centrifugation. Protein concentrations were determined with a Nanodrop 2000 Spectrophotometer (Thermo Scientific). About 15–30 µg of protein extracts were supplemented with 1% 2-mercaptoethanol (Sigma-Aldrich) and 0.0005% Bromophenol blue (Sigma-Aldrich), incubated at 95 °C for 5 min, separated in 6, 12, or 14% polyacrylamide gels, and transferred to nitrocellulose membranes (Amersham Protran 0.45 NC, Cytiva) using a Trans-Blot SD Semi-Dry Transfer Cell apparatus (Bio-Rad). Primary antibodies were: a rabbit polyclonal anti-PC4 (same as above, 1:5000 dilution), a rabbit polyclonal anti-RPA32 pSer33 (same as above, 1:2000 dilution), a rabbit polyclonal anti-RPA32 (Bethyl Laboratories, A300-244A; 1:2000 dilution), a mouse monoclonal anti-beta Actin (Abcam, ab8224; 1:5000 dilution), a rabbit polyclonal anti-PML (same as above, 1:1000 dilution), a rabbit polyclonal anti-TRF2 (Novus Biologicals, NB110-57130; 1:1000 dilution), a rabbit monoclonal anti-Vinculin (Cell Signaling Technology, #13901; 1:1000 dilution), a rabbit polyclonal anti-Bloom (Bethyl Laboratories, A300-110A; 1:1000 dilution), a rabbit polyclonal anti-ATRX (Bethyl Laboratories, A301-045A; 1:1000 dilution), a rabbit polyclonal anti-RNA PolII (Bethyl Laboratories, A300-653A; 1:2000 dilution), a rabbit polyclonal anti-phospho RNA PolII (S2) (Bethyl Laboratories, A300-654A; 1:10000 dilution), a rabbit polyclonal anti-phospho RNA PolII (S5) (Bethyl Laboratories, A300-655A; 1:10000 dilution), and a goat polyclonal anti-Helic2 (Santa Cruz Biotechnology, SC-68563; 1:1000 dilution). Secondary antibodies were HRP-conjugated goat anti-mouse and anti-rabbit IgGs (Novus Biologicals, NB7539 and NB7160, respectively; 1:3000 dilution). Signal detection was done using the ECL detection reagents (Cytiva) and an Amersham 680 RGB Imager (Cytiva).

## RNA and DNA preparation and analysis

Total RNA was prepared using the TRIzol reagent (Invitrogen) or the Nucleospin RNA kit (Macherey-Nagel) and treated twice with DNaseI (Qiagen). Poly(A) + RNA was purified using Oligo d(T)25 Magnetic Beads (New England Biolabs). For northern blotting, RNA was separated in 1.2% agarose gels containing 0.7% formaldehyde, blotted onto nylon membranes (Amersham Hybond™ -N+, Cytiva) by capillary transfer and hybridized with a radiolabeled telomeric probe detecting UUAGGG (Telo2 probe; (Azzalin et al, 2007)) at 55 °C overnight. Post-hybridization washes were twice in 2x SSC (0.3 M NaCl, 0.03 M trisodium citrate (VWR), pH 7.2), 0.2% SDS for 20 min, and once in 0.2x SSC, 0.2% SDS for 30 min at 55 °C. After signal detection membranes were stripped and re-hybridized overnight at 50 °C with oligonucleotides detecting beta Actin (5′-GTGAGGATCTTCATGAGGTAGTCAGT-CAGGT-3′), U6 (5′-GGAACGCTTCACGAATTTGCGT-3′), 18S rRNA (5′-CCATCCAATCGGTAGTAGCG-3′), or the 18S rRNA precursor ITS1 (5′-CCTCGCCCTCCGGGCTCCGTTAATTGATC-3′), 5′-end

labeled with T4 polynucleotide kinase (New England Biolabs) and [γ-32P]ATP (PerkinElmer). Post-hybridization washes were twice in 2× SSC, 0.2% SDS for 20 min and once in 1× SSC, 0.2% SDS for 30 min at 50 °C for beta Actin and U6, or twice in 2× SSC, 0.5% SDS and once in 0.5× SSC, 0.5% SDS all for 30 min at 50 °C for 18S and ITS1. For dot-blot hybridization, 2 μg of RNA were transferred onto nylon membranes and hybridized as above for Telo2. After signal detection, membranes were stripped and re-hybridized as above to detect the 18S rRNA. For C-circle assays, genomic DNA was isolated by phenol:chloroform extraction (Sigma-Aldrich) and treated with 40 μg/ml RNaseA, followed by ethanol precipitation. Then, 150–500 ng of DNA were incubated with 7.5 U phi29 (Φ29) DNA polymerase (New England Biolabs) in the presence of dATP, dTTP, and dGTP (1 mM each; Thermo Scientific) at 30 °C for 8 h, followed by heat-inactivation at 65 °C for 20 min. Amplification products were dot-blotted onto nylon membranes and hybridized with a radiolabeled Telo2 probe as above. Radioactive signals were detected using an Amersham Typhoon IP imager (Cytiva) and quantified using ImageJ software.

## RNA FISH

Cells grown on coverslips were rinsed with ice-cold 1× PBS and incubated for 7 min with ice with CSK buffer supplemented with 10 mM Vanadyl Ribonucleoside Complex (New England Biolabs). Cells were washed once with ice-cold 1× PBS, fixed with 4% formaldehyde in 1x PBS for 10 min at room temperature, washed twice with 1× PBS for 5 min at room temperature, and incubated again with CSK buffer for 10 min at room temperature. Cells were incubated with 70, 90, and 100% ethanol for 5 min each at room temperature and air-dried. A Telo2 DNA probe, strand-specifically labeled by random priming with Cy3-dCTP (PerkinElmer), dTTP and dATP, was resuspended in RNA hybridization solution (50% formamide, 10% dextran sulfate (Sigma-Aldrich), 2 mg/ml BSA, 2× SSC, 0.3 M NaCl, 0.03 M trisodium citrate, pH 7.2), denatured at 75 °C for 5 min and applied onto slides. Slides were incubated overnight at 37 °C in a light-protected humid chamber. Cells were washed as follows for 5 min at each step: three times with 2× SSC/50% formamide at 39 °C, three times with 2× SSC at 39 °C, and once with 2× SSC. DNA was counterstained with 100 ng/ml DAPI in 2× SSC for 5 min at room temperature, and coverslips were mounted on slides in Vectashield. Images were acquired with a Zeiss Cell Observer equipped with a cooled Axiocam 506 m camera and a 63X/1.4NA oil DIC M27 PlanApo N objective. Quantification of nuclear TERRA intensity and TERRA foci was performed using semiautomated ImageJ macros on "sum slices" Z-projections of the image stacks, converted to 8-bit images. DAPI staining, after threshold setting and background subtraction, was used to define the nuclear boundaries (minimum size of 50 pixels). Nuclear TERRA intensity was measured as the raw integrated density of signal inside the pre-determined nuclear boundaries in the TERRA channel, after background subtraction. For quantification of the total number of foci, the "Find Maxima" function was applied to the TERRA channel to count the number of single points with a prominence of 5 within the defined nuclear boundaries.

## Chromatin immunoprecipitation (ChIP)

Cells were harvested from plates by scraping, centrifuged at 500×g at 4 °C for 5 min, and resuspended in 1% formaldehyde in 1× PBS

for 30 min at room temperature. After quenching with 125 mM glycine for 5 min, cells were washed three times with 1x PBS by centrifuging at 800×g for 5 min. Cross-linked cells were lysed by resuspension in ChIP lysis buffer (1% SDS, 10 mM EDTA (Sigma-Aldrich), 50 mM Tris-HCl pH 8), supplemented with cOmplete Protease Inhibitor Cocktail (Roche), sonicated two to three times using a Bioruptor apparatus (Diagenode) at 4 °C (settings: 30 s "ON"/30 s "OFF"; power: "High"; time: 15 min), and centrifuged at 16,000×g for 10 min at 4 °C. About 1 mg of lysate was diluted in ChIP dilution buffer (1% Triton X-100, 20 mM Tris-HCl pH 8, 2 mM EDTA pH 8, 150 mM NaCl) to a final volume of 1 ml and precleared by incubation with 50 μl of protein A/G-Agarose beads (Santa Cruz Biotechnology) or 20 μl of Dynabeads Protein G (Invitrogen), previously blocked with sonicated E.coli genomic DNA and BSA (New England Biolabs). Extracts were centrifuged at 800×g at 4 °C for 5 min and then incubated with 1 μg of rabbit anti-PC4 antibody or with 3 μg of mouse monoclonal anti-Myc antibody (Cell Signaling Technology) for 3 h at 4 °C on a rotating wheel. Immunocomplexes were isolated by incubation with Protein A/G-Agarose beads or Dynabeads Protein G at 4 °C overnight on a rotating wheel. Beads were washed four times with ChIP wash buffer 1 (0.1% SDS, 1% Triton X-100, 2 mM EDTA pH 8, 150 mM NaCl, 20 mM Tris-HCl pH 8) and once with ChIP wash buffer 2 (0.1% SDS, 1% Triton X-100, 2 mM EDTA pH 8, 500 mM NaCl, 20 mM Tris-HCl pH 8) with centrifugation steps at 800×g at 4 °C for 5 min or magnetic separation. Beads were incubated with ChIP elution buffer (1% SDS, 100 mM NaHCO3 (Sigma-Aldrich)) containing 40 μg/ml RNaseA for 1 h at 37 °C, followed by incubation at 65 °C overnight to reverse crosslinks. DNA was purified using the Wizard SV gel and PCR Clean-up system (Promega), denatured at 95 °C for 10 min, dot-blotted onto a nylon membrane, and hybridized at 55 °C with a radiolabeled Telo2 probe or at 50 °C with a radiolabeled G-rich telomeric oligonucleotide (TelG; 5′-TTAGGGTTAGGGTTAGGGTTAGGGTTAGGG-3′). Post-hybridization washes for Telo2 were as above, and for TelG were twice in 2× SSC, 0.5% SDS for 20 min and once in 0.5× SSC, 0.5% SDS for 30 min at 50 °C. After signal detection, membranes were stripped and re-hybridized overnight with radiolabeled Alu-repeat oligonucleotides (5′-GTGATCCGCCCGCCTCGGCCTCC-CAAAGTG-3′) at 50 °C. Post-hybridization washes were twice in 2× SSC, 0.2% SDS for 20 min and once in 0.5 × SSC, 0.2% SDS for 30 min at 50 °C. Radioactive signals were detected using an Amersham Typhoon IP imager and quantified using ImageJ software.

## His-tagged protein purification

PC4 cDNAs, wild type or W89A, carrying a N-terminal 6x His-tag sequence, were cloned into a pET-Duet1 plasmid (Sigma-Aldrich) and transformed into competent BL21 cells (Thermo Scientific). Cells were grown in 5 ml of LB medium overnight at 37 °C and 500 μl of those cultures were then inoculated into 100 ml of fresh LB media. After growing cells at 37 °C for 4 h, protein expression was induced with 100 μM IPTG (PanReac) for 2 h at 30 °C. Cells were collected by centrifugation (4200 × g for 10 min at 4 °C) and lysed by sonication in 20 mL of HEX buffer (20 mM HEPES-NaOH pH 7.5 (Sigma-Aldrich), 0.2% Triton X-100, 80 mM NaCl, 0. 5 mM PMSF (Roche), 10% glycerol) supplemented with His-tag protease inhibitors (Sigma-Aldrich). Samples were sonicated twice using an

MSE Soniprep 150 apparatus (Sanyo) on ice (settings: 3 min "ON"/ 2 min "OFF"; power: "Maximum") and centrifuged at 4200×*g* for 20 min at 4 °C. Supernatants were incubated with an agarose resin charged with divalent cobalt (Thermo Scientific) on gravity-flow polypropylene columns (Thermo Scientific). Beads were washed three times with 5 ml of HEX buffer supplemented with increasing concentrations of imidazole (10, 25, and 50 mM; PanReac). Bound proteins were eluted in 1 ml of HEX buffer supplemented with 250 mM imidazole. Protein concentration and purity were determined using the Bradford reagent (NZYtech) and BSA reference samples, followed by fractionation in polyacrylamide gels and staining with BlueSafe reagent (NZYtech).

### Electrophoretic mobility shift assay (EMSA)

5-repeats oligonucleotides were purchased from Integrated DNA Technologies. The sequences of the telomeric oligonucleotides were: 5′-(TTAGGG)5-3′, 5′-(TTACGG)5-3′, 5′-(CCCTAA)5-3′; the sequence of the unrelated oligonucleotide was 5′-GGACTC-TAGCGTGGATCCTTAAGCTAGAAT-3′. Oligonucleotides were 5′-end labeled with T4 polynucleotide kinase and [γ32P]-ATP and then purified using the Oligo Clean and Concentrator kit (Zymo Research). Recombinant protein and oligonucleotides were incubated with 20 μl of EMSA buffer (50 mM HEPES pH 8, 1 mM DTT (Thermo Scientific), 100 mM NaCl, 0.01% BSA, 2% glycerol) for 20 min on ice followed by 10 min at 25 °C. After incubation, 4 μl of 6× gel loading buffer (30% glycerol, 0.3% bromophenol blue) were added to the reactions, and samples were separated by electrophoresis in native 8% polyacrylamide gels in pre-cooled 0.5× TBE. Gels were run at 120 V for 1 h at 4 °C and dried. Radioactive signals were detected using an Amersham Typhoon IP imager and quantified using ImageJ software.

### Statistical analysis

For large datasets ($n > 50$), we assessed the normality of the distributions using the Shapiro–Wilk test and found that nearly all datasets did not fit a normal distribution. Therefore, we employed a non-parametric two-tailed Mann–Whitney *U*-test. For small datasets ($n = 3$), given that the non-parametric test lacks sufficient power, we assumed normal distribution and calculated statistical significance using the parametric unpaired two-tailed Student's *t*-test. All tests were performed using GraphPad Prism. Significant (<0.05) or nearly significant (>0.05 and <0.055) *P* values are indicated in the figures and figure legends. No blinding approach was applied to the analyses.

## Data availability

This study does not include data deposited in external repositories. Data supporting the findings of this work are available within the paper. Materials can be requested from the corresponding author.

The source data of this paper are collected in the following database record: biostudies:S-SCDT-10_1038-S44319-024-00295-3.

## Peer review information

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

## Acknowledgements

We thank Bruno Silva for precious help during the project, Alexandre Maréchal, Jean-Christophe Dubois, and Raymund J. Wellinger (Université de Sherbrooke, Canada) for sharing unpublished data and for fruitful discussions, Joana Rodrigues for providing ImageJ macros, and Massimo Lopes, Bruno Fuchs, Arturo Londoño-Vallejo, Ulrike Kutay, Markus Lobrich, Joachim Lingner, and Didier Trono for sharing reagents. We also thank the Bioimaging and Flow Cytometry facilities of the GIMM for their invaluable services. This work was supported by Portuguese national funds through FCT – Fundação para a Ciência e a Tecnologia, I.P. (project 2021.00143.CEECIND to CMA; project PTDC/MED-ONC/7864/2020 to PLA and CMA), by LaCaixa Foundation (project LCF/PR/HP21/52310016 to CMA), and by TessellateBIO. SS is supported by a PhD fellowship from FCT (2022.11369.BD).

## Author contributions

**Sara Salgado**: Conceptualization; Data curation; Formal analysis; Investigation; Writing—original draft; Writing—review and editing. **Patricia L Abreu**: Conceptualization; Data curation; Formal analysis; Supervision; Investigation; Writing—original draft; Project administration; Writing—review and editing. **Beatriz Moleirinho**: Data curation; Formal analysis; Investigation. **Daniela S Guedes**: Data curation; Formal analysis; Investigation. **Lee Larcombe**: Conceptualization; Data curation; Formal analysis; Writing—original draft. **Claus M Azzalin**: Conceptualization; Data curation; Formal analysis; Supervision; Funding acquisition; Investigation; Writing—original draft; Project administration; Writing—review and editing.

Source data underlying figure panels in this paper may have individual authorship assigned. Where available, figure panel/source data authorship is listed in the following database record: biostudies:S-SCDT-10_1038-S44319-024-00295-3.

## Disclosure and competing interests statement

CMA is a founder and shareholder of TessellateBIO. LL consults for TessellateBIO.

# Expanded View Figures

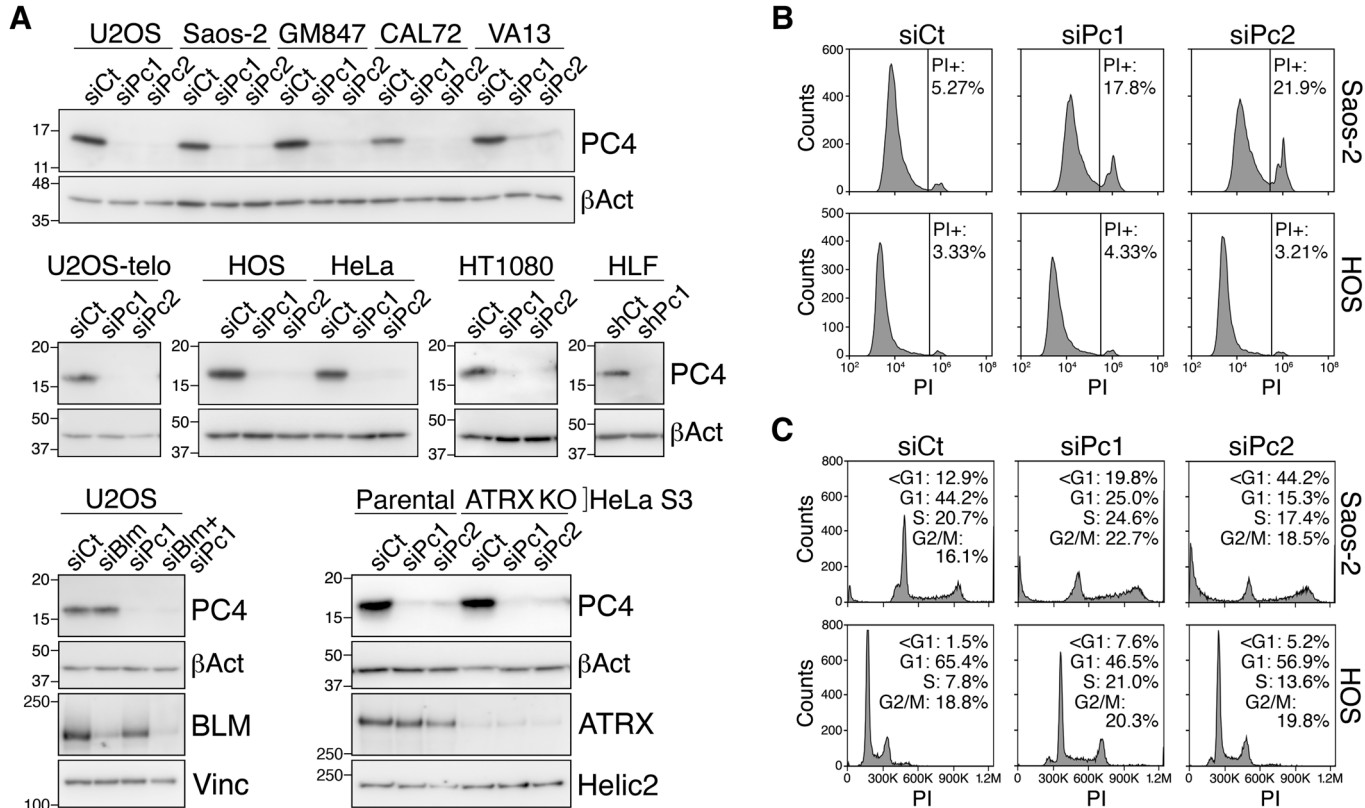

**Figure EV1. PC4 depletion induces death in ALT cells.**

(**A**) Western blot analysis of PC4, BLM, and ATRX protein levels. Cells were transfected with siRNAs or infected with shRNA lentiviruses, and total proteins were extracted 72 h after siRNA transfections or 5 days after shRNA infections. Beta Actin (βAct), Vinculin (Vinc), and Helic2 serve as loading controls. Marker molecular weights are on the left of the gels in kDa. (**B**) Examples of FACS profiles of cells stained with PI without permeabilization. The indicated cell lines were depleted of PC4 for 9 days. Cell counts (y-axis) are plotted against PI intensity (x-axis) for one representative experiment. Numbers are percentages of cells positive to PI staining as defined by the indicated gates (vertical bars). (**C**) Examples of FACS profiles of Saos-2 and HOS cells ethanol-fixed and stained with PI. Cells were transfected with siRNAs every 72 h for a total of 9 and 15 days, respectively. Cell counts (y-axis) are plotted against PI intensity (x-axis). The percentages of cells with different DNA contents, including sub-G1 (<G1), are indicated.

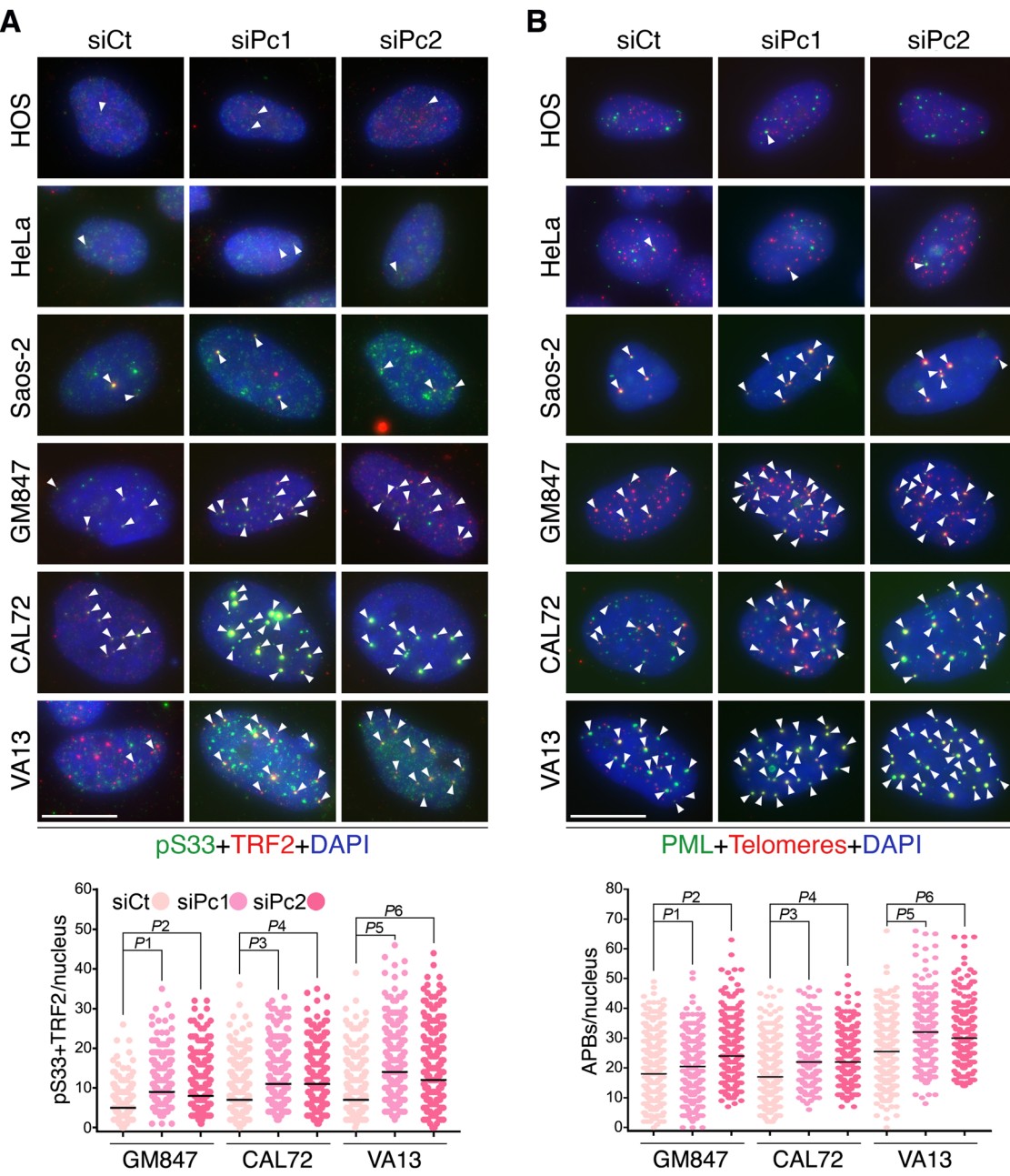

**Figure EV2. PC4 restricts telomeric replication stress and APBs.**

Examples of pS33 (green) and TRF2 (red) double IF (**A**) and of PML IF (green) combined with telomeric DNA FISH (red; (**B**)) in the indicated cell lines transfected with siRNAs and harvested 72 h after transfection. DAPI-stained DNA is in blue. Arrowheads point to co-localization events. The plots at the bottom show the co-localization events per nucleus. At least 100 nuclei were analyzed for each sample in each of the three biological replicates. For the plots in (**A**): $P1 < 0.0001$; $P2 < 0.0001$; $P3 < 0.0001$; $P4 < 0.0001$; $P5 < 0.0001$; $P6 < 0.0001$. For the plots in (**B**): $P1 = 0.001$; $P2 < 0.0001$; $P3 < 0.0001$; $P4 < 0.0001$; $P5 < 0.0001$; $P6 < 0.0001$ (Mann–Whitney $U$-test). Scale bars: 10 μm.

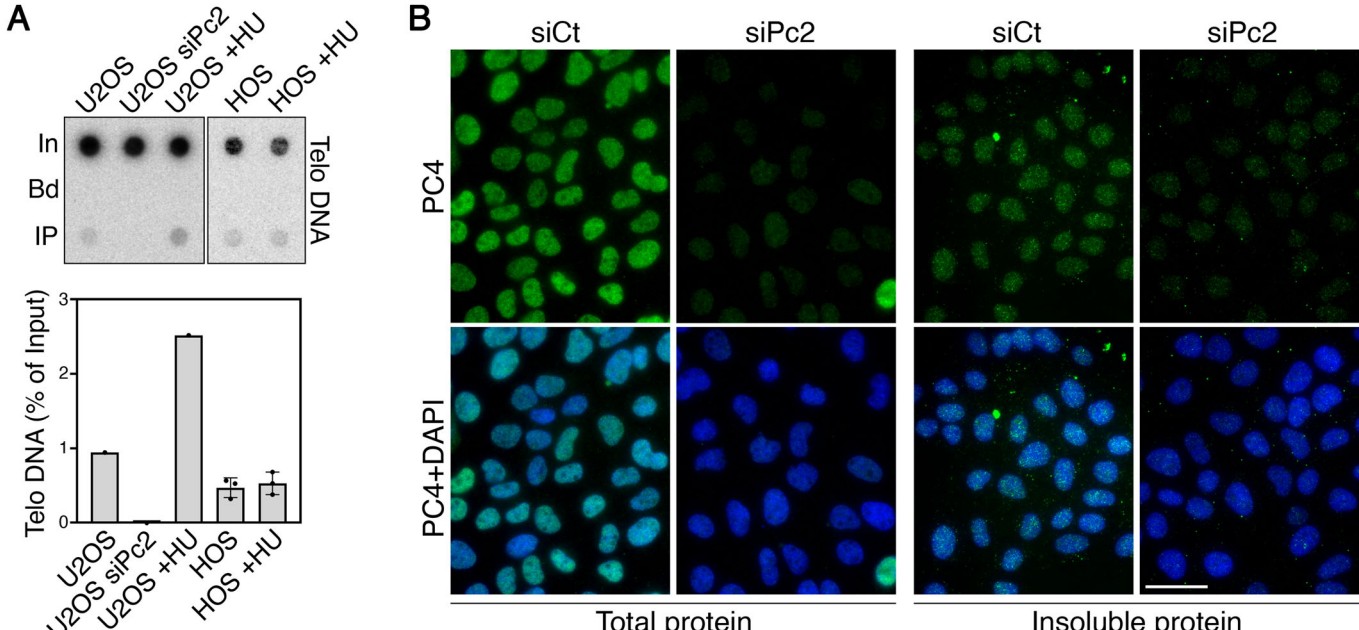

**Figure EV3. PC4 is a nuclear protein, partly bound to chromatin regions including telomeres.**

(A) Dot-blot hybridization of endogenous PC4 ChIPs in U2OS or HOS cells using radiolabeled probes to detect telomeric (Telo) DNA. Cells were either transfected with siPc2 and harvested 72 h after transfection, or treated with 0.2 mM hydroxyurea (HU) for 16 h. In: Input (1%), Bd: only beads control (50%), IP: PC4 immunoprecipitation (50%). Signals were quantified and graphed (bottom) as the fraction of input DNA found in the corresponding IP samples, after subtraction of Bd-associated signals. For U2OS, results are from one biological replicate. For HOS, bars and error bars are means and SDs from three biological replicates. The disappearance of telomeric DNA signal in the IP fraction of PC4-depleted U2OS cells confirms the specificity of the antibody. (B) Examples of PC4 (green) immunostaining in U2OS cells transfected with siPc2 or siCt and harvested 72 h after transfection. Cells were either permeabilized with mild detergent prior to fixation (right panels) or left untreated (left panels), in order to visualize chromatin-bound PC4 and total PC4, respectively. The substantial decrease in staining in PC4-depleted cells confirms the specificity of the antibody. Scale bar: 30 µm.

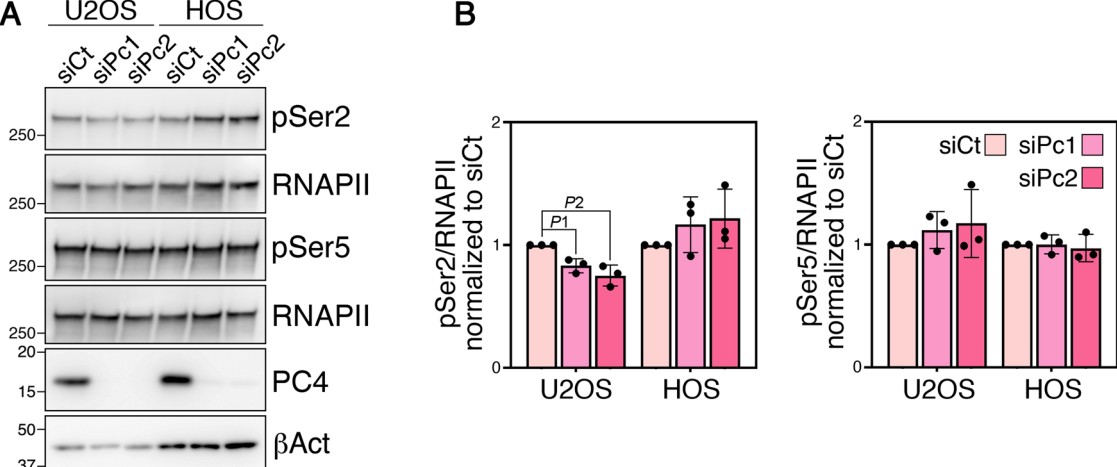

**Figure EV4.  Effects of PC4 depletion on RNAPII phosphorylation.**

(A) Western blot analysis of PC4, RNAPII phosphorylated at CTD Serine 2 (pSer2) and 5 (pSer5), and total RNAPII. Total proteins were extracted 72 h after siRNA transfection. Beta Actin (βAct) serves as a loading control. Marker molecular weights are on the left of the gels in kDa. (B) pSer2 and pSer5 signals were quantified and graphed after normalization using the corresponding total RNAPII signals. siCt values are set to 1. Bars and error bars are means and SDs from three biological replicates. $P1 = 0.0071$; $P2 = 0.0077$ (Student's $t$-test). Source data are available online for this figure.

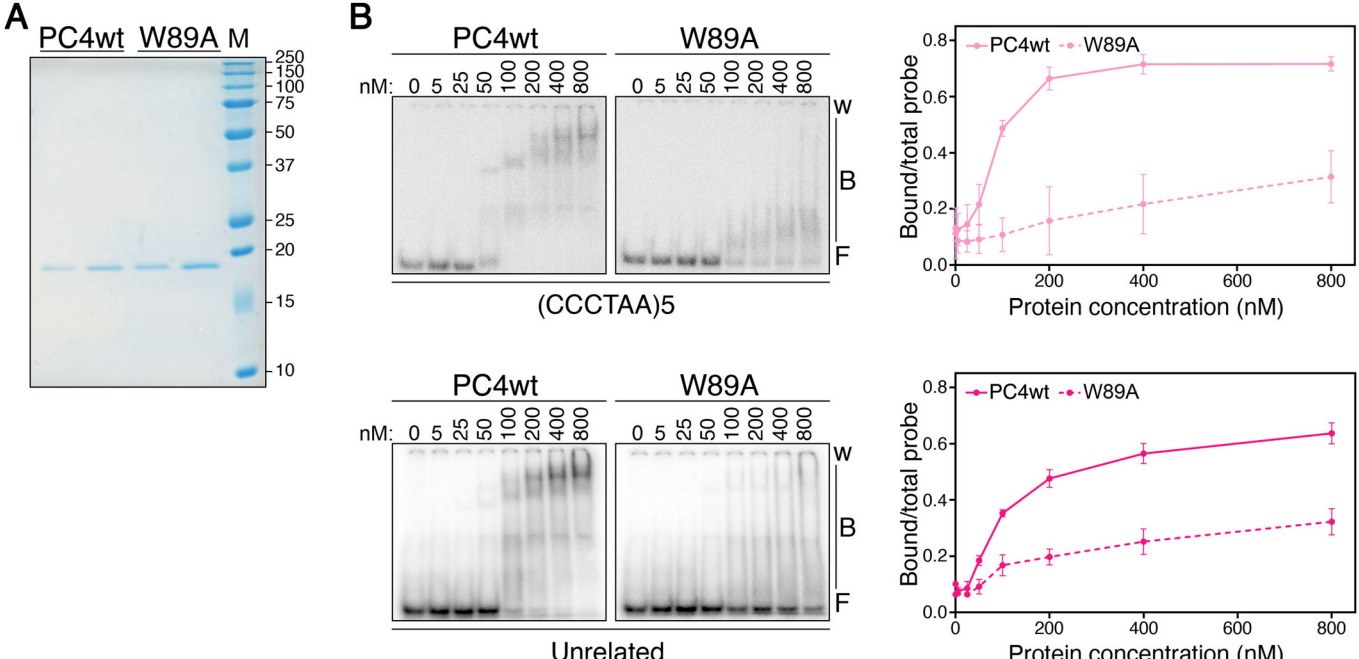

**Figure EV5. The ssDNA binding-deficient mutant W89A binds to ssDNA substrates.**

(A) 150 and 300 ng of PC4wt and 250 and 500 ng of W89A recombinant proteins were size-fractionated by SDS-PAGE and stained with BlueSafe reagent. Marker (M) molecular weights are on the right in kDa. (B) Electrophoretic mobility shift assay was performed with recombinant PC4wt and W89A proteins and the indicated ssDNA oligonucleotides. w: wells; B: bound probe; F: free probe. The graphs show quantifications of bound oligonucleotides graphed as a fraction of the total signal within each lane. Data points and error bars are means and SDs from three biological replicates.

