## [Peer Review File · EMBO Reports]

Human PC4 supports telomere stability and viability in cells utilizing the alternative lengthening of telomeres mechanism

Sara Salgado, Patricia Lona Abreu, Beatriz Moleirinho, Daniela Guedes, Lee Larcombe, and Claus Azzalin

Corresponding author(s): Claus Azzalin (cmazzalin@medicina.ulisboa.pt)

Review Timeline:

Submission Date:	12th Apr 24
Editorial Decision:	7th May 24
Revision Received:	6th Sep 24
Editorial Decision:	25th Sep 24
Revision Received:	7th Oct 24
Accepted:	11th Oct 24

Editor: Esther Schnapp

Transaction Report:

Dear Claus,

Thank you for the submission of your manuscript to EMBO reports. We have now received the full set of referee reports that is pasted below.

As you will see, all referees acknowledge that the findings are interesting. They also have a few suggestions for how the study could be strengthened, and I think all should be addressed, except, that may be not all comments by referee 3 on Fig 3 need to be addressed. Please let me know if you have any comments. We can discuss the exact revision requirements also in a video chat, if you like.

I would thus like to invite you to revise your manuscript with the understanding that the referee concerns must be fully addressed and their suggestions taken on board. Please address all referee concerns in a complete point-by-point response. Acceptance of the manuscript will depend on a positive outcome of a second round of review. It is EMBO reports policy to allow a single round of major revision only and acceptance or rejection of the manuscript will therefore depend on the completeness of your responses included in the next, final version of the manuscript.

We realize that it is difficult to revise to a specific deadline. In the interest of protecting the conceptual advance provided by the work, we recommend a revision within 3 months (7th Aug 2024). Please discuss the revision progress ahead of this time with the editor if you require more time to complete the revisions.

- 1) A data availability section providing access to data deposited in public databases is missing. If you have not deposited any data, please add a sentence to the data availability section that explains that.
- 2) Your manuscript contains statistics and error bars based on $n=2$. Please use scatter blots in these cases. No statistics should be calculated if $n=2$.

3) We replaced Supplementary Information with Expanded View (EV) Figures and Tables that are collapsible/expandable online. A maximum of 5 EV Figures can be typeset. EV Figures should be cited as 'Figure EV1, Figure EV2' etc... in the text and their respective legends should be included in the main text after the legends of regular figures.

5) a complete author checklist, which you can download from our author guidelines . Please insert information in the checklist that is also reflected in the manuscript. The completed author checklist will also be part of the RPF.

6) Please note that all corresponding authors are required to supply an ORCID ID for their name upon submission of a revised manuscript (). Please find instructions on how to link your ORCID ID to your account in our manuscript tracking system in our Author guidelines

7) Before submitting your revision, primary datasets produced in this study need to be deposited in an appropriate public database (see <https://www.embopress.org/page/journal/14693178/authorguide#datadeposition>). Please remember to provide a reviewer password if the datasets are not yet public. The accession numbers and database should be listed in a formal "Data Availability" section placed after Materials & Method (see also <https://www.embopress.org/page/journal/14693178/authorguide#datadeposition>). Please note that the Data Availability Section is restricted to new primary data that are part of this study. * Note - All links should resolve to a page where the data can be accessed. *
If your study has not produced novel datasets, please mention this fact in the Data Availability Section.

- the name of the statistical test used to generate error bars and P values,
- the number (n) of independent experiments (please specify technical or biological replicates) underlying each data point,
- the nature of the bars and error bars (s.d., s.e.m.),
- If the data are obtained from n {less than or equal to} 2, use scatter blots showing the individual data points.

I look forward to seeing a revised form of your manuscript when it is ready.

Best,

Esther

Referee #1:

In this manuscript entitled "Human PC4 supports telomere stability and viability in cells utilizing the alternative lengthening of telomeres mechanism", Salgado et al. show that the Positive Cofactor 4 (PC4) (aka SUB1) is essential for the viability of Alternative Lengthening of Telomeres (ALT) cells, mainly through its suppressive role on telomere instability and ALT activity. To my knowledge, they are the first to report this.

Overall, this paper is well written, the objectives are clearly stated, and the findings are supported by a variety of experiments. However, some of the conclusions drawn by the authors could benefit from additional experiments or further discussion of limitations in their study.

1: The authors state that they have identified 14 cell lines that use the ALT mechanism but do not describe the basis for this selection.

2: More than a third of the selected ALT cell lines are osteosarcomas and, the five ALT-positive cell lines with the highest dependency on PC4 are derived from bone cancers. Given this selection bias toward the bone lineage, it is possible that PC4 is essential for the proliferation of bone cancers rather than ALT. This seems supported by previous data identifying PC4/SUB1 as a "supertarget" for osteosarcoma cells (see Chetverina et al., 2023 <https://doi.org/10.3390/cancers15113042>).

3: Figure 1C shows that depletion of PC4 leads to decreased cell numbers in ALT cell lines. To a lesser extent, this is also true for non-ALT cell lines, especially with the treatment with siPc1. The authors should address whether the cell quantity is also more reduced in HeLa with siPc1 treatment. The authors also use the term "significantly" to describe this reduced effect of PC4, but there is no statistical test to support this claim.

4: Figure 1D shows that ALT cells die more from PC4 depletion than non-ALT cells. This, taken together with the data from Figure 1C and 1E, leads the authors to conclude that the decrease observed in non-ALT cells is likely due to a decrease in proliferation. This hypothesis could be easily confirmed by labeling the cells with a non-toxic dye diluting at each division such as DiD/Dil/DiO, and checking the level of fluorescence, either by microscopy or flow cytometry.

5: The authors do not show that the sensitivity of ALT-positive cells is ALT-specific. They could test this by co-depleting factors that are required for alt activity (i.e. BLM or PML KO).

6: Figure 2A: The authors claim that "PC4 depletion increased pS33 telomeric localization in all ALT cells". While colocalization events of stress markers at telomeres are higher in siPC4-treated cells, it appears that stress markers appear to increase globally in ALT cells. Knowing this, it is possible that the increased localization at telomeres is the result of this global increase. To substantiate the conclusion that pS33 increases at restricted genomic loci, it may be relevant to quantify pS33 foci in the whole nucleus, not only at telomeres.

7: Since PC4 is involved in RNAPII regulation and TERRA molecules are transcribed by RNAPII, PC4 may influence TERRA expression levels. The authors investigate the possible involvement of TERRA in the telomeric defects observed in PC4-depleted cells. They rule out the role of deregulated TERRA at the global level, but they do not address the possibility of the increased localization of TERRA at telomeres.

8: Interestingly, when applying statistics to a reduced number of data (histograms, n=3), the authors chose to use a parametric Student's t-test. In contrast, they use a non-parametric Mann-Whitney U test when comparing samples with more than 100 values. This is not inherently wrong, but could the authors justify this choice?

9: In Figure 3, the numbering is missing (e.g. A, B, C).

Referee #2:

This is an interesting manuscript showing that PC4 has a role in alternative lengthening of telomeres. The results are quite interesting and I would favour publication in EMBO Reports.

I have a scientific question. Both FANCM and PC4 are top hits from the DepMap analysis (Fig. 1A). Are these two genes in the

same pathway? Can the authors co-deplete FANCM and PC4 by siRNA and see the effect in the experiment of Fig. 1D and in the experiment of Fig. 2A or 2B or 2C? The results would be interesting, no matter what the outcome is.

Referee #3:

This manuscript by Salgado et al. identifies the RNAPII cofactor PC4, as a transcriptional regulator that responds to damaged telomeric DNA and is essential for (ALT) cancer cell survival. Previous studies have shown that PC4 dynamically regulates transcription by RNAPII in a dose-dependent manner and can protect cell viability from drug-induced replication stress. Here, the authors report that cells that utilize ALT telomere lengthening were selectively sensitive to PC4 depletion compared to cancer cell lines that use telomerase for telomere lengthening. In the absence of PC4, telomeric replication stress (TRS) increased, as did ALT activity, suggesting that PC4 restrains ALT-induced TRS. Interestingly, the ssDNA binding domain of PC4 appears to be dispensable for this role, prompting the question of the mechanism of PC4 suppression of ALT TRS. Overall, the presented studies convincingly show the dependency of cancer cells that use ALT on PC4. The data is well presented for the most part. The mechanistic aspects of the study fall short of providing insights of how PC4 contributes to ALT. Some additional clarity could be provided regarding how PC4 regulates ALT.

Major Comments (in order of figures):

Fig 1. Can the authors assess PC depletion's effect on normal cells (untransformed) and some additional telomerase-expressing lines? These are a bit limited in the paper. Related to this, ATRX or DAXX loss can correlate with ALT. Is there any relationship between ATRX or DAXX mutation and PC4 dependency? Could this be addressed in silico or experimentally?

Fig 2A. The authors compare telomeric replicative stress between ALT and telomerase-positive cell lines and conclude that PC4 depletion suppresses TRS only in ALT cells. However, there is a significant increase in pS33 at telomeres in the HeLa telomerase-positive cells. The authors describe it as a mild increase. Is it a comparable increase, given the low starting levels of TRS in HOS or HeLa? What would a fold change look like here?

Fig. 2D. Please conduct a triplicate experiment in Saos2 to derive a standard deviation.

Fig. 3C. FANCM loss leads to R-loops. Does PC4 loss induce R-loops? Does PC4 bind to R-loops? Is PC association with telomeres RNAPII dependent? Test effect of DRB or other RNAPII inhibitors? Is there any effect on RNAPII phosphorylation by PC4 siRNA?

Fig. 4B-D. With the ssDNA being dispensable for PC4-related activities at telomeres the studies in Fig.4. seem inconclusive. Are other regions of PC4 required? What about the 1-62 deletion? PC4 also contains a lysine-rich domain. Could PARP activity be required for PC4 localization? PC4 also contains a SAEC motif and can be phosphorylated by TFII subunits of RNAPII. Can any of these be ruled out or confirmed as essential in regulating its function or recruitment?

Fig. EV4. There was no appreciable change in TERRA levels, so it is unlikely that PC4 regulates ALT through telomeric transcription. Does PC4 depletion alter TERRA localization to the telomere or TERRA stability? In addition, PC4 also relies on transcriptional activity for localization. As PC4 does not affect TERRA transcription, it may be downstream. Is TERRA required for PC4 localization to the telomere (relates to fig 3 also).

Minor Comments:

1. In the introduction - define shelterin to help those unfamiliar with what that is.
2. In the introduction - it would be helpful to provide information on any other RNAPII-related factors that have been implicated in ALT telomere lengthening. Or is PC4 unique (an outlier)?
3. For the imaging-related data in graphs throughout the paper, there are no SDs or SEMs. Shouldn't these be presented?
4. Fig.3. there are no figure labels (A, B, etc.).
5. Fig.3. Can co-localizations in Control and telomerase-expressing cells be indicated? Looks like there are some.
6. Fig.3. In DNA in IP (%) is a confusing axis label. % of Input would be clearer.
7. Fig. EV3. PC4 binding to telomeres is increased in the presence of HU treatment in U2OS cells. Does this also happen in response to DSBs or replication stress in telomerase-positive cells? Literature citation is sufficient.
8. Fig.EV5B. Has it been previously shown that the W89A mutant retains some level of ssDNA binding? Maybe I missed a citation. If not, can this difference be explained?

Point-by-point response to the Referees' comments.

We would like to thank the three anonymous Referees for their appreciation of our work and constructive criticisms. In response to the comments raised, we conducted an extensive series of experiments, which we believe have significantly strengthened the manuscript compared to its initial version. Unfortunately, as detailed below, some of the suggested experiments yielded negative results. While we have included several new experiments in the revised manuscript, we have opted to leave some of the negative results only in this rebuttal, as we feel they do not substantially contribute to the main conclusions of our study. Please note that all changes in the text are highlighted in red. Here is our detailed point-by-point response to each of the Referees' comments (in bold).

Referee #1:

In this manuscript entitled "Human PC4 supports telomere stability and viability in cells utilizing the alternative lengthening of telomeres mechanism", Salgado et al. show that the Positive Cofactor 4 (PC4) (aka SUB1) is essential for the viability of Alternative Lengthening of Telomeres (ALT) cells, mainly through its suppressive role on telomere instability and ALT activity. To my knowledge, they are the first to report this. Overall, this paper is well written, the objectives are clearly stated, and the findings are supported by a variety of experiments. However, some of the conclusions drawn by the authors could benefit from additional experiments or further discussion of limitations in their study.

We thank the Referee for their supportive comments and constructive criticisms. We can confirm that our study is the first to report on the functions of PC4 at human telomeres.

1: The authors state that they have identified 14 cell lines that use the ALT mechanism but do not describe the basis for this selection.

ALT cells were identified through an extensive review of the literature. We now indicate this in the text (page 3) and cite the relevant articles that were utilized.

2: More than a third of the selected ALT cell lines are osteosarcomas and, the five ALT-positive cell lines with the highest dependency on PC4 are derived from bone cancers. Given this selection bias toward the bone lineage, it is possible that PC4 is essential for the proliferation of bone cancers rather than ALT. This seems supported by previous data identifying PC4/SUB1 as a "supertarget" for osteosarcoma cells (see Chetverina et al., 2023 <https://doi.org/10.3390/cancers15113042>).

We thank the Referee for directing us to the study by Chetverina et al., which we now reference in the manuscript. We believe that the authors' conclusion that PC4 is a "supertarget" for osteosarcoma can be interpreted differently. Out of the 13 osteosarcoma cell lines screened at DepMap, only 5 (approximately 40%, consistent with the prevalence of ALT across osteosarcomas; MacKenzie et al. 2021, *Cancers* **13**: 2384) show significant sensitivity to PC4 depletion (PC4 Gene KO

effect < 0.5). Of these 5 lines, 4 (U2OS, Saos-2, CAL72, and NOS-1) are known to be ALT-positive, and the fifth (OS052) is likely to be as it was isolated from a juvenile osteosarcoma. Therefore, we propose that PC4 specifically supports the viability of osteosarcomas with an activated ALT mechanism rather than osteosarcomas in general. This interpretation is now discussed in the text on page 3.

3: Figure 1C shows that depletion of PC4 leads to decreased cell numbers in ALT cell lines. To a lesser extent, this is also true for non-ALT cell lines, especially with the treatment with siPc1. The authors should address whether the cell quantity is also more reduced in HeLa with siPc1 treatment. The authors also use the term "significantly" to describe this reduced effect of PC4, but there is no statistical test to support this claim.

We conducted PC4 depletion experiments in HeLa cells using siPc1 and incorporated the results into Fig. 1C. In this case, we did not observe any decrease in cell numbers over the 13-day experiment. Additionally, we included an analysis of cell numbers and viability using another telomerase-positive cell line, HT1080 (fibrosarcoma), where PC4 depletion did not cause negative effects both on cell numbers and viability (Fig. 1C and D). These data (discussed on page 4) further support the hypothesis that PC4 plays a more critical role in maintaining the viability of ALT cells rather than telomerase-positive cells. Regarding the statistical analysis, we acknowledge that no statistical tests were shown for Fig. 1C due to the large volume of data points. We have removed the term "significantly" from the text describing Fig. 1C and included all statistical tests in the accompanying Data Source files.

4: Figure 1D shows that ALT cells die more from PC4 depletion than non-ALT cells. This, taken together with the data from Figure 1C and 1E, leads the authors to conclude that the decrease observed in non-ALT cells is likely due to a decrease in proliferation. This hypothesis could be easily confirmed by labeling the cells with a non-toxic dye diluting at each division such as DiD/Dil/DiO, and checking the level of fluorescence, either by microscopy or flow cytometry.

While we appreciate this suggestion, we decided not to include these additional experiments. We believe that the cell viability data presented in Fig. 1D clearly demonstrates that PC4 depletion causes more death in ALT than telomerase positive cells. To avoid any reference to proliferation rates for any of the cell lines, we have rephrased the relevant text (page 4). We do not rule out the possibility that PC4 could also influence the proliferation rates of cells (potentially both ALT and telomerase-positive), and we plan to address this point in a future study.

5: The authors do not show that the sensitivity of ALT-positive cells is ALT-specific. They could test this by co-depleting factors that are required for alt activity (i.e. BLM or PML KO).

This is an excellent point, and we thank the Referee for the insightful suggestion. In response, we performed double depletion of PC4 and the Bloom helicase (BLM) in U2OS cells and assessed cell viability using PI staining of unfixed cells followed by FACS analysis. As illustrated in Fig. 1D, approximately 46% of cells are PI-permeable in cells depleted only of PC4, compared to 22% in cells

co-depleted of PC4 and BLM. This, along with the observation that telomerase expression in U2OS cells does not rescue their sensitivity to PC4 depletion, strongly supports the idea that PC4 is essential for the viability of cells that rely on an activated ALT mechanism. This finding is discussed in detail on pages 4 and 5.

6: Figure 2A: The authors claim that "PC4 depletion increased pS33 telomeric localization in all ALT cells". While colocalization events of stress markers at telomeres are higher in siPC4-treated cells, it appears that stress markers appear to increase globally in ALT cells. Knowing this, it is possible that the increased localization at telomeres is the result of this global increase. To substantiate the conclusion that pS33 increases at restricted genomic loci, it may be relevant to quantify pS33 foci in the whole nucleus, not only at telomeres.

We agree with the Referee that, upon PC4 depletion in ALT cells, there appears to be a general increase in pS33 foci, some of which do not colocalize with the telomeric marker TRF2 (Fig. 2A and Fig. 3A). To measure this, we quantified the total number of defined pS33 foci in U2OS and CAL72 cells and indeed found a slight tendency towards accumulation of nuclei with increased pS33 foci in siPc1- and siPc2-transfected cells (Fig. R1). It is important to note that, despite this accumulation, the pS33 staining remains focal rather than pan-nuclear (Fig. 2A and Fig. 3A). Additionally, western blot analysis of total pS33 does not show a significant induction of the marker (Fig. 2E), suggesting that pS33 does not accumulate extensively throughout the genome but rather at specific loci, including telomeres and other regions of unknown nature. We have clarified this point in the text (page 6).

Fig. R1: Analysis of the effects of PC4 depletion on total pS33 accumulation on chromatin. Quantification of the number of pS33 foci in U2OS and CAL72 cells transfected with the indicated siRNAs and harvested 72 h after transfection. Cells are the same as in Figures 2A and 3A. Quantifications were performed using semiautomated ImageJ macros on “sum slices” Z-projections of the image stacks, converted to 8-bit images. DAPI staining, after threshold setting and background subtraction, was used to define the nuclear boundaries (minimum size of 50 pixels). The “Find Maxima” function was applied to the pS33 channel to count the number of single points with a prominence of 25 within the defined nuclear boundaries. Each dot represents one nucleus, black bars are medians. At least 100 nuclei were analyzed for each sample in each of 3 independent experiments. *P* values (Mann-Whitney U test) are: *P*1<0.0001; *P*2=0.5680; *P*3=0.5147; *P*4<0.0001.

7: Since PC4 is involved in RNAPII regulation and TERRA molecules are transcribed by RNAPII, PC4 may influence TERRA expression levels. The authors investigate the possible involvement of

TERRA in the telomeric defects observed in PC4-depleted cells. They rule out the role of deregulated TERRA at the global level, but they do not address the possibility of the increased localization of TERRA at telomeres.

We conducted TERRA FISH experiments on U2OS cells depleted of PC4. The analysis of both TERRA intensity and foci number did not reveal any significant differences between depleted and control cells (Fig. EV4D), suggesting that deregulated TERRA localization to telomeres is unlikely to be responsible for the telomeric defects observed in PC4-depleted cells. Additionally, in response to one of Referee #3's major comments on Fig. EV4 (see below), we now show that PC4 depletion in U2OS cells does not affect TERRA degradation rates (Fig. EV4B). This further supports the conclusion that PC4 is not primarily involved in regulating TERRA biogenesis. These data are discussed on page 7.

8: Interestingly, when applying statistics to a reduced number of data (histograms, n=3), the authors chose to use a parametric Student's t-test. In contrast, they use a non-parametric Mann-Whitney U test when comparing samples with more than 100 values. This is not inherently wrong, but could the authors justify this choice?

For large datasets (n>50), we assessed the normality of the distributions using the Shapiro–Wilk test in GraphPad Prism and found that nearly all datasets do not fit a normal distribution. Therefore, following the guidelines for authors of EMBO Reports, we opted to use the non-parametric Mann-Whitney U test. For datasets with n=3, given that the non-parametric test lacks sufficient power, we assumed normal distribution and calculated statistical significance using the Student's unpaired t-test. This is now explained in the Materials and Methods section (page 16).

9: In Figure 3, the numbering is missing (e.g. A, B, C).

We erroneously uploaded a version of the figure where the panel label layer was rendered invisible. We corrected this.

Referee #2:

This is an interesting manuscript showing that PC4 has a role in alternative lengthening of telomeres. The results are quite interesting and I would favor publication in EMBO Reports. I have a scientific question. Both FANCM and PC4 are top hits from the DepMap analysis (Fig. 1A). Are these two genes in the same pathway? Can the authors co-deplete FANCM and PC4 by siRNA and see the effect in the experiment of Fig. 1D and in the experiment of Fig. 2A or 2B or 2C? The results would be interesting, no matter what the outcome is.

We thank the Referee for their supportive comments and for recognizing the significance of our work. Regarding the Referee's scientific question, addressing the genetic interactions between FANCM and PC4 would be challenging. The effects of FANCM depletion on ALT cell viability and telomere stability are dramatically stronger and appear earlier than those induced by PC4 depletion

(Pan et al. 2017, *Proc Natl Acad Sci USA* **114**: E5940; Silva et al. 2019, *Nat Commun* **10**: 2253; Lu et al. 2019, *Nat Commun* **10**: 2252). Additionally, the inevitable incomplete protein depletion associated to the use of siRNAs, rather than knockout cells (which cannot be generated given the essentiality of both PC4 and FANCM), would make the interpretation of the results particularly complex.

Nonetheless, we performed double depletions of FANCM and PC4 in U2OS cells using siPc1 and an siRNA against FANCM (siFM; Fig. R2) that results in partial depletion of the protein, thereby mitigating the severe effects on cell proliferation and viability. Three days after transfection, the fraction of dead cells in PC4-depleted and FANCM-depleted samples was 10.7% and 16.4%, respectively. In the double-depleted sample, this fraction increased to 27.7%. Similarly, the fraction of cells in the G2/M phase, a characteristic feature associated with FANCM depletion, was 12.4%, 26.4%, and 57.8% in PC4-depleted, FANCM-depleted, and double-depleted cells, respectively (Fig. R2). These experiments may suggest that FANCM and PC4 support ALT cell proliferation and viability through parallel pathways. However, it is also possible that the two proteins function within the same pathway, with PC4 depletion amplifying the effects induced by partial FANCM depletion. Given the complexity and ambiguity of these results, we believe it is best to exclude them from the manuscript.

Fig. R2: Analysis of the effects of PC4 and FANCM double depletion on cell viability and cell cycle progression. (A) Western blot analysis of PC4 and FANCM protein levels in U2OS cells transfected with siPc1 (to deplete PC4), siFM (to deplete FANCM), or a control siRNA (siCt). Total proteins were extracted 72 h after siRNA transfection. Beta Actin (β Act) and Vinculin (Vinc) were used as loading controls. Marker molecular weights are indicated on the left in kDa. (B) Cells as in A were stained with propidium iodide (PI) without permeabilization and analyzed by FACS. The numbers indicate the percentage of cells positive for PI staining (dead/dying cells) as defined by the indicated gates (vertical lines). (C) Cells as in A were fixed with ethanol, PI-stained, and analyzed by FACS. The numbers represent the percentages of cells in each indicated phase of the cell cycle. Note that in our previous work (Silva et al. 2019, *Nat Commun* **10**: 2253), we demonstrated that more than 50% of U2OS cells depleted of FANCM using a more efficient siRNA become PI-permeable and accumulate in G2/M within 72 hours after siRNA transfection.

Referee #3:

This manuscript by Salgado et al. identifies the RNAPII cofactor PC4, as a transcriptional regulator that responds to damaged telomeric DNA and is essential for (ALT) cancer cell survival. Previous studies have shown that PC4 dynamically regulates transcription by RNAPII in a dose-dependent manner and can protect cell viability from drug-induced replication stress. Here, the authors report that cells that utilize ALT telomere lengthening were selectively sensitive to PC4 depletion compared to cancer cell lines that use telomerase for telomere lengthening. In the absence of PC4, telomeric replication stress (TRS) increased, as did ALT activity, suggesting that PC4 restrains ALT-induced TRS. Interestingly, the ssDNA binding domain of PC4 appears to be dispensable for this role, prompting the question of the mechanism of PC4 suppression of ALT TRS. Overall, the presented studies convincingly show the dependency of cancer cells that use ALT on PC4. The data is well presented for the most part. The mechanistic aspects of the study fall short of providing insights of how PC4 contributes to ALT. Some additional clarity could be provided regarding how PC4 regulates ALT.

We thank the Referee for their supportive comments and very constructive criticisms.

Major Comments (in order of figures):

Fig 1. Can the authors assess PC depletion's effect on normal cells (untransformed) and some additional telomerase-expressing lines? These are a bit limited in the paper. Related to this, ATRX or DAXX loss can correlate with ALT. Is there any relationship between ATRX or DAXX mutation and PC4 dependency? Could this be addressed in silico or experimentally?

We conducted cell counting and viability assays using an additional telomerase-positive cell line (HT1080, fibrosarcoma) and primary lung fibroblasts (HLFs). The data are presented in Fig. 1C and D and Fig. EV1A and discussed on page 4. Our analysis indicates that HT1080 cells are insensitive to PC4 depletion, further confirming that telomerase-positive cells are less dependent on PC4 for survival. Interestingly, we found that HLFs also exhibit sensitivity to PC4 depletion, although to a slightly lesser extent than ALT cells. Specifically, approximately 30% of PC4-depleted HLFs remain on the plate at day 15 of depletion compared to control cells. Moreover, viability assays show a 1.5-fold increase in dead cells upon PC4 depletion. These findings suggest that PC4 might be necessary to support the viability of some primary, non-transformed cells, possibly challenging the safety of targeting PC4 for ALT cancer treatment. As a consequence, we toned down all statements regarding clinical applications. Nonetheless, we believe that the main conclusion of our work — that PC4 supports the viability of cells with an active ALT mechanism — remains unchallenged.

We also addressed the possible synthetic interaction between PC4 and ATRX inactivation, a particularly relevant suggestion given that all ALT cells used in this study lack ATRX expression. We depleted PC4 in an ATRX-knockout (KO) HeLa S3 clone, as well as in parental HeLa S3 cells and measured PI-permeable dead cells. Unlike parental HeLa cells, the HeLa S3 clone exhibited a 1.5-fold increase in dead cells upon PC4 depletion (Fig. 1D; Fig. EV1A), possibly due to an unknown genetic alteration. Nonetheless, ATRX-KO cells behaved identically to parental ones (Fig. 1D; Fig. EV1A). We conclude that ATRX loss does not render cells dependent on PC4 for survival. These findings are discussed on pages 4 and 5.

Fig 2A. The authors compare telomeric replicative stress between ALT and telomerase-positive cell lines and conclude that PC4 depletion suppresses TRS only in ALT cells. However, there is a significant increase in pS33 at telomeres in the HeLa telomerase-positive cells. The authors describe it as a mild increase. Is it a comparable increase, given the low starting levels of TRS in HOS or HeLa? What would a fold change look like here?

We agree with the Referee that there is an increase in telomeric pSer33 foci in telomerase-positive HeLa cells as well. The table below presents the fold change (FC) in the median number of telomeric pSer33 foci in PC4-depleted cells compared to siRNA control-transfected cells, using the same data as for Fig. 2A and Fig. 3A. While we do not see a median fold increase in HOS cells, we could not calculate (not determined, ND) the fold change for HeLa cells because the median number of foci in siCt-transfected cells is 0. However, the median increases to 1 in both PC4-depleted samples.

siRNA	U2OS			Saos-2			GM847			CAL72			VA13			HOS			HeLa		
	Ct	Pc1	Pc2	Ct	Pc1	Pc2	Ct	Pc1	Pc2	Ct	Pc1	Pc2	Ct	Pc1	Pc62	Ct	Pc1	Pc2	Ct	Pc1	Pc2
Median	2	4	4	2	3	3	5	9	8	7	11	11	7	14	12	1	1	1	0	1	1
FC vs siCt	1	2.000	2.000	1	1.500	1.500	1	1.800	1.600	1	1.571	1.571	1	2.000	1.714	1	1.000	1.000	ND	ND	ND

Although we understand the rationale behind considering fold increases, we believe that one or few telomeric pSer33 foci per cell are unlikely to have a significant impact on cell physiology, consistent with the fact that we do not observe viability defects in PC4-depleted HeLa cells. This is why we initially stated: “A much milder accumulation of telomeric pSer33 was also observed in PC4-depleted HeLa and HOS cells”. To avoid any confusion, we now write: “PC4 depletion increased telomeric pSer33 levels in HeLa and HOS cells as well, but the total number of co-localization events remained consistently lower than in ALT samples (Fig. 2A; Fig. 3A)”. This sentence is found on page 5.

Fig. 2D. Please conduct a triplicate experiment in Saos2 to derive a standard deviation.

We performed an additional c-circle measurement for Saos-2 cells and included it in Fig. 2D.

Fig. 3C. FANCM loss leads to R-loops. Does PC4 loss induce R-loops? Does PC4 bind to R-loops? Is PC association with telomeres RNAPII dependent? Test effect of DRB or other RNAPII inhibitors? Is there any effect on RNAPII phosphorylation by PC4 siRNA?

and

Fig. 4B-D. With the ssDNA being dispensable for PC4-related activities at telomeres the studies in Fig.4. seem inconclusive. Are other regions of PC4 required? What about the 1-62 deletion? PC4 also contains a lysine-rich domain. Could PARP activity be required for PC4 localization? PC4 also contains a SAEC motif and can be phosphorylated by TFII subunits of RNAPII. Can any of these be ruled out or confirmed as essential in regulating its function or recruitment?

To address some of the points raised regarding Figures 3 and 4B-D, we conducted several additional experiments. First, we depleted PC4 in U2OS and HOS cells and assessed RNAPII phosphorylation via western blotting. We used antibodies specific for phosphorylated CTD Serine 2 and Serine 5, and normalized the obtained values against total RNAPII. Signal quantification showed only a minor decrease in pSer2 levels in PC4-depleted U2OS cells (see Fig. EV3 and text on page 7).

Furthermore, we treated U2OS cells with various small molecule inhibitors, including Triptolide (RNPII degrader), DRB (inhibitor of RNAPII CTD phosphorylation), TBB (CK2 inhibitor), KU55933 (ATM inhibitor), VE821 (ATR inhibitor), and Olaparib (PARP inhibitor). We then performed ChIP assays using antibodies against endogenous PC4. Treatments with Triptolide, DRB, TBB, and Olaparib did not significantly affect PC4 binding to telomeres when compared to the control treatment with solvent (DMSO) (Fig. R3). This suggests that RNAPII, CK2, and PARP do not play a role in PC4 recruitment to telomeres. Treatments with KU55933 and VE821 reduced PC4 binding to telomeric DNA by 25% and 14%, respectively, indicating a possible minor involvement of ATM and ATR in this process (Fig. R3). Given the negligible or minor effects observed with the tested small molecule inhibitors, we decided not to include these data in the manuscript. However, we plan to further explore the potential connections between PC4, ATM and ATR in future studies.

Fig. R3: Analysis of the effects of inhibition of RNAPII, CK2, ATM, ATR and PARP on PC4 binding to telomeres. Dot-blot hybridization of endogenous PC4 ChIP samples from U2OS cells treated with Triptolide (1 μ M for 3 h), 5,6-dichloro-1-beta-D-ribofuranosylbenzimidazole (DRB; 100 μ M for 3 h), 4,5,6,7-tetrabromobenzotriazole (TBB; 50 μ M for 5 h), KU55933 (10 μ M for 5 h), VE821 (10 μ M for 5 h), and Olaparib (20 μ M for 5 h). All inhibitors were dissolved in DMSO. The dot-blot membrane was hybridized using a radiolabeled telomeric probe. In: Input (1%), Bd: beads-only control (50%), IP: PC4 immunoprecipitation (50%). Signals were quantified and graphed (bottom) as the fraction of input DNA present in the corresponding IP samples, after subtraction of bead-associated signals. Two independent experiments were performed.

Fig. EV4. There was no appreciable change in TERRA levels, so it is unlikely that PC4 regulates ALT through telomeric transcription. Does PC4 depletion alter TERRA localization to the telomere or TERRA stability? In addition, PC4 also relies on transcriptional activity for localization. As PC4 does not affect TERRA transcription, it may be downstream. Is TERRA required for PC4 localization to the telomere (relates to fig 3 also).

We conducted TERRA FISH experiments on U2OS cells depleted of PC4 and did not observe significant alterations in either TERRA intensity or foci number (Fig. EV4D). Please refer to our

response to Referee #1's point 7 for a more detailed explanation. Additionally, we measured TERRA stability in U2OS cells transfected with siCt and siPc1 by performing northern blotting of total RNA extracted from cells treated with the global transcription inhibitor Actinomycin D over a 10-hour time course. Although we were unable to reach TERRA half-life due to the onset of cell death after 10 hours, TERRA decay rates were similar in both control and PC4-depleted cells (Fig. EV4B). These experiments further suggest that deregulated TERRA is not responsible for the telomeric defects observed following PC4 depletion in ALT cells. These data are discussed on page 7.

To test whether TERRA could recruit PC4 to telomeres, we employed two previously developed U2OS cell lines, vp30 and sid4, where TERRA transcription can be up- or down-regulated by the expression of Transcription Activator-like Effectors (TERRA-TALEs) that recognize TERRA promoters and are fused to either an RNAPII transcription activator (VP64) or suppressor (sid4X) module, respectively (Silva et al., 2019, *Nat Commun* **10**: 2253; Silva et al., 2022, *Proc Natl Acad Sci USA* **119**: e2208669119). We also included a control clone (nls1) expressing TERRA-TALEs not fused with any transcription regulator. TERRA-TALEs were induced by treating cells with doxycycline (dox), and PC4 binding to telomeres was evaluated by ChIP using antibodies against the endogenous protein. We observed a decrease in PC4 binding in dox-treated sid4 cells, while no change was detected in dox-treated vp30 cells (Fig. R4). However, dox treatment in nls1 control cells also led to a decrease in PC4 binding to telomeres, likely due to secondary effects exerted by dox (Fig. R4). Although these data could suggest that TERRA might support PC4 recruitment to telomeres, the effect exerted by dox is confounding and additional and different experimental approaches need to be used before reaching a solid conclusion.

Fig. R4: Analysis of the effects of TERRA transcription on PC4 binding to telomeres. (A) Western blot analysis of TERRA-TALE expression in the indicated U2OS-derived clonal cell lines. TERRA-TALEs are C-terminally tagged with an HA epitope and were detected using anti-HA antibodies. Prior to total protein extraction, cells were treated with 50 ng/ml dox for 48 h. Proteins stained with Ponceau S serve as a loading control. Marker molecular weights are indicated on the left in kDa. (B) Dot-blot hybridization of endogenous PC4 ChIP samples from cells as in A. In: Input (2%), Bd: beads-only control (50%), IP: PC4 immunoprecipitation (50%). Signals were quantified and indicated at the bottom as the fraction of input DNA present in the corresponding IP samples, after subtraction of bead-associated signals. One representative experiment is shown.

Minor Comments:

1. In the introduction - define shelterin to help those unfamiliar with what that is.

We now write that shelterin is a multiprotein complex comprising the two telomeric dsDNA binding proteins TRF1 and TRF2, as well as Rap1, TIN2, TPP1 and POT1 (page 2).

2. In the introduction - it would be helpful to provide information on any other RNAPII-related factors that have been implicated in ALT telomere lengthening. Or is PC4 unique (an outlier)?

To our knowledge, this is the first report of an RNAPII cofactor involved in ALT telomere stability. More broadly, there is work suggesting that RNAPII-mediated transcription deregulation, achieved, for example, through inactivating the chromatin remodeler ATRX or the RNaseH1 endoribonuclease, might impact ALT telomere maintenance. However, this hypothesis has never been tested and the effects would likely be indirect rather than directly linked to RNAPII regulation. Additionally, we do not know whether the functions of PC4 in ALT cells are linked to its role as an RNAPII-related factor or whether they are independent, as proposed for the functions exerted by PC4 in maintaining genome stability in cells experiencing replication stress (see introduction).

3. For the imaging-related data in graphs throughout the paper, there are no SDs or SEMs. Shouldn't these be presented?

As explained in response to Referee #1's point 8, for imaging-related data (large datasets) we tested the normality of the distributions and verified that most of the datasets do not meet the assumptions of the test. Thus, for the graphical representation we opted for showing all the data points and the median as measure of central tendency, instead of the mean and the associated measures of dispersion.

4. Fig.3. there are no figure labels (A, B, etc.).

We erroneously uploaded a figure where the layer of the panel labels was rendered invisible. We have corrected this.

5. Fig.3. Can co-localizations in Control and telomerase-expressing cells be indicated? Looks like there are some.

We indicated co-localization events in U2OS siCt and telomerase-positive cells in Fig. 4B (previously Fig. 3B). Please note that, for consistency, we now indicate co-localization events also in all nuclei shown in Fig. 3 (previously Fig. EV2).

6. Fig.3. In DNA in IP (%) is a confusing axis label. % of Input would be clearer.

We now write: "DNA (% of Input)".

7. Fig. EV3. PC4 binding to telomeres is increased in the presence of HU treatment in U2OS cells. Does this also happen in response to DSBs or replication stress in telomerase-positive cells? Literature citation is sufficient.

We could not find any reports of PC4 binding to telomeres in telomerase-positive cells experiencing telomeric DNA damage or replication stress. Indeed, we believe we are the first to investigate in details the functions of PC4 at telomeres. To address this point, we treated HOS cells with hydroxyurea (HU) and performed ChIP assays using antibodies against PC4 (data shown in Fig.

EV2A). In contrast to what we observed in U2OS cells, the HU treatment did not increase PC4 binding to telomeres in HOS cells, suggesting that replications stress leads to PC4 recruitment to telomeres specifically in ALT cells. These findings are reported on page 6.

8. Fig.EV5B. Has it been previously shown that the W89A mutant retains some level of ssDNA binding? Maybe I missed a citation. If not, can this difference be explained?

According to Werten et al. (*EMBO J* **17**: 5103-5111, 1998), the W89A mutant completely lacks ssDNA binding activity. This is why we state in the text, “However, unexpectedly, the W89A protein retained the ability to form complexes with all oligonucleotides” (page 8). Although we do not currently understand the reason for this discrepancy, we find this finding relevant and believe it is certainly worth reporting.

Dear Claus,

Thank you for the submission of your revised manuscript. We have now received the enclosed reports from the referees and I am happy to say that both support its publication now. Only a few editorial requests will need to be addressed before we can proceed with the official acceptance of your manuscript:

- Figures EV3A and EV4C: There are similarities in the blots that we would like to check against the source data (SD). Please submit the SD for these panels at the next stage.
- The Disclosure statement needs to be moved to after the Acknowledgements.
- The reference format needs to be corrected to the EMBO reports style (as Harvard style).
- Fig. 3A is called out before Fig. 2B, Fig. 3B called out before Fig. 2D, please correct.
- The Methods section should include a separate Reagents and Tools Table file (listing key reagents, experimental models, software and relevant equipment and including their sources and relevant identifiers). A downloadable template (.docx) for the Reagents and Tools Table can be found in our author guidelines: <<https://www.embopress.org/page/journal/14693178/authorguide#manuscriptpreparation>>
- Please note that the exact p values are not provided in the legends of figures 2a-c; 3a-b; 5b.
- Please indicate the statistical test used for data analysis in the legend of figure 1a.
- Please note that information related to n is missing in the legends of figures 1a-b.
- Please note that the white arrowheads are not defined in the legend of figure 3a-b. This needs to be rectified.

EMBO press papers are accompanied online by A) a short (1-2 sentences) summary of the findings and their significance, B) 2-3 bullet points highlighting key results and C) a synopsis image that is exactly 550 pixels wide and 200-600 pixels high (the height is variable). The synopsis image should provide a sketch of the major findings, like a graphical abstract. Please note that text needs to be readable at the final size. Please send us this information along with the final manuscript.

Referee #1:

I commend the authors for their thoughtful and detailed revision of this manuscript. They have carefully addressed all of my comments and conducted additional experiments to resolve the key points raised. The revised manuscript is significantly stronger and has addressed all major concerns. I believe it will be of great interest to the telomere biology community

Referee #3:

Thank you to the authors for taking all 3 reviewers critiques on. Lovely work.

The authors have addressed all minor editorial requests.

Dr. Claus Azzalin
Gulbenkian Institute for Molecular Medicine (GIMM)
Av. Professor Egas Moniz
Lisbon 1649-028
Portugal

Dear Claus,

I am very pleased to accept your manuscript for publication in the next available issue of EMBO reports. Thank you for your contribution to our journal.

The synopsis image you sent is fine but was not sent at the correct size of 550 pixels wide. When I resize your image the text is slightly blurred. Could you may be send us a new image file at the correct size ?

Best,
Esther
